

# Neogene paleogeography provides context for understanding the origin and spatial distribution of cryptic diversity in a widespread Balkan freshwater amphipod

Michał Grabowski[1], Tomasz Mamos[1], Karolina Bącela-Spychalska[1], Tomasz Rewicz[2] and Remi A. Wattier[3]

[1] Department of Invertebrate Zoology and Hydrobiology, University of Lodz, Łódź, Poland
[2] Laboratory of Microscopic Imaging and Specialized Biological Techniques, University of Lodz, Łódź, Poland
[3] Laboratoire Biogéosciences, Université de Bourgogne Franche-Comté, UMR CNRS 6282, Dijon, France

## ABSTRACT

**Background**. The Balkans are a major worldwide biodiversity and endemism hotspot. Among the freshwater biota, amphipods are known for their high cryptic diversity. However, little is known about the temporal and paleogeographic aspects of their evolutionary history. We used paleogeography as a framework for understanding the onset of diversification in *Gammarus roeselii*: (1) we hypothesised that, given the high number of isolated waterbodies in the Balkans, the species is characterised by high level of cryptic diversity, even on a local scale; (2) the long geological history of the region might promote pre-Pleistocene divergence between lineages; (3) given that *G. roeselii* thrives both in lakes and rivers, its evolutionary history could be linked to the Balkan Neogene paleolake system; (4) we inspected whether the Pleistocene decline of hydrological networks could have any impact on the diversification of *G. roeselii*.
**Material and Methods**. DNA was extracted from 177 individuals collected from 26 sites all over Balkans. All individuals were amplified for ca. 650 bp long fragment of the mtDNA cytochrome oxidase subunit I (COI). After defining molecular operational taxonomic units (MOTU) based on COI, 50 individuals were amplified for ca. 900 bp long fragment of the nuclear 28S rDNA. Molecular diversity, divergence, differentiation and historical demography based on COI sequences were estimated for each MOTU. The relative frequency, geographic distribution and molecular divergence between COI haplotypes were presented as a median-joining network. COI was used also to reconstruct time-calibrated phylogeny with Bayesian inference. Probabilities of ancestors' occurrence in riverine or lacustrine habitats, as well their possible geographic locations, were estimated with the Bayesian method. A Neighbour Joining tree was constructed to illustrate the phylogenetic relationships between 28S rDNA haplotypes.
**Results**. We revealed that *G. roeselii* includes at least 13 cryptic species or molecular operational taxonomic units (MOTUs), mostly of Miocene origin. A substantial Pleistocene diversification within-MOTUs was observed in several cases. We evidenced secondary contacts between very divergent MOTUs and introgression of nDNA. The Miocene ancestors could live in either lacustrine or riverine habitats yet their presumed geographic localisations overlapped with those of the Neogene lakes. Several extant riverine populations had Pleistocene lacustrine ancestors.

Corresponding author
Michał Grabowski,
michalg@biol.uni.lodz.pl

**Discussion.** Neogene divergence of lineages resulting in substantial cryptic diversity may be a common phenomenon in extant freshwater benthic crustaceans occupying areas that were not glaciated during the Pleistocene. Evolution of *G. roeselii* could be associated with gradual deterioration of the paleolakes. The within-MOTU diversification might be driven by fragmentation of river systems during the Pleistocene. Extant ancient lakes could serve as local microrefugia during that time.

## INTRODUCTION

Due to its turbulent geological history over the last 30 million years, the Balkan Peninsula has been characterized by enormous and dynamic landscape change, resulting in great complexity. Both paleoenvironmental reconstructions and present day physiographic patterns are showing high habitat diversity and patchiness for both terrestrial and aquatic ecosystems (*Popov et al., 2004*). Movement of the African tectonic plate towards the north, and its collision with Eurasian plate in Neogene, at ca. 20 million years ago (Ma), caused the Tethys regression and uplift of the Hellenic and Dinaric Mountains, followed by abrupt paleoclimatic changes in Plio-Pleistocene that was intensified by Milankovitch climatic oscillations (*Nisancioglu, 2010*). As a result, the formation of hydrological networks throughout the Balkan Peninsula was considered a very complex process, with temporal existence of large lake systems, fragmentation of drainages, frequent river captures and progressing karstification. Thus, at present, the hydrological network is composed of several larger, and hundreds of smaller, drainages with relatively unstable watersheds (*Economou et al., 2007*). The isolation of habitat patches and emergence of new habitats in the course of the formation of the Balkan Peninsula provided a stimulus for speciation events, including: vicariance; peripatry; and parapatry (e.g., *Griffiths, Kryštufek & Reed, 2004*; *McInerney et al., 2014*; *Mamos et al., 2016*). In consequence, the region is recognized as one of the most important present day hot-spots of biodiversity and endemism globally, and a model system for studies upon biogeography and the evolution of numerous organisms (*Blondel & Médail, 2009*; *Blondel et al., 2010*; *Poulakakis et al., 2015*). Conventional faunistic and floristic studies show that the Balkans house ca. 35% of the Palearctic species and more than 6% of the world's freshwater species, with at least 43% of species being local endemics (*Tierno de Figueroa et al., 2013*). In addition, the area has been the focus of numerous late Pleistocene phylogeographic studies. Many of these studies highlight that it was probably a key glacial refugium for a number of animal and plant species during the last glacial period, and a source of post glacial recolonisation (*Hewitt, 2000*; *Tzedakis, 2004*; *Pauls, Lumbsch & Haase, 2006*). In addition, some studies also pointed out the presence of cryptic diversity in various biota (*Mamos et al., 2014*; *Previšić et al., 2014*; *Sworobowicz et al., 2015*). In several cases, divergent lineages at the molecular level (e.g., >2% divergence for the mtDNA COI gene) are recognized within one morphospecies (*Bickford et al., 2007*; *Milankov et al., 2008*;

*Klobučar et al., 2013*), while some studies documented cryptic diversity using DNA barcoding, via delimiting Molecular Operation Taxonomic Units (MOTU) (*Copilaş-Ciocianu & Petrusek, 2015*; *Sworobowicz et al., 2015*), and others addressed the evolutionary history behind the observed diversity patterns (*Wysocka et al., 2013*; *Wysocka et al., 2014*; *Mamos et al., 2016*). Several papers point out that the lineage divergence, in at least some species inhabiting the Balkan Peninsula, dates back to the Neogene (20.03–2.58 Ma) or to the early Pleistocene (2.58–0.78 Ma); a time-span well before the Last Glacial Maximum (*Parmakelis et al., 2006*; *Copilaş-Ciocianu & Petrusek, 2015*). Particularly in the case of freshwater animals, pre-Pleistocene changes in watersheds and isolation of drainages associated with mountain chain uplift have probably shaped the phylogeography more profoundly than the Pleistocene glaciations and associated climatic shifts (e.g., *Verovnik, Sket & Trontelj, 2005*; *Mamos et al., 2016*). However, so far there have been relatively few attempts to link the distribution patterns of extant aquatic taxa to particular geological events in the Balkans (*Veith, Kosuch & Vences, 2003*; *Fromhage, Vences & Veith, 2004*; *Falniowski & Szarowska, 2011*; *Mamos et al., 2016*). This is not surprising, given that the hydrological system of the area has been restructured many times during the already mentioned turbulent and ca. 30 My long paleogeographical history of the area, which makes the interpretation of results quite a difficult task.

Gammaridean amphipods (Crustacea) are an ideal model group to study diversity and distribution patterns in freshwater systems. They are strictly aquatic with very limited abilities to disperse among isolated water basins via ectozoochory (e.g., *Segerstråle, 1954*; *Rachalewski et al., 2013*). Also, they are usually unable to survive periods of drought longer than a few days in any stage of their ontogeny (*Väinölä et al., 2008*; *Bącela-Spychalska et al., 2013*; *Rachalewski et al., 2013*). Amphipods play a key role in maintaining energy flow in freshwater ecosystems, decomposing organic matter (*Graca et al., 2001*; *Piscart et al., 2009*) and, with their great abundance, providing an important food source for fish (*Macneil, Dick & Elwood, 1999*; *Padovani et al., 2012*). Being quite sensitive to pollution, native species of *Gammarus* Fabricius, 1775 are used as standard organisms for ecotoxicology testing (e.g., *Sroda & Cossu-Leguille, 2011*; *Gerhardt, Bloor & Mills, 2011*). Substantial cryptic diversity has already been detected in several groups of freshwaters amphipods in water bodies such as: isolated spring systems in desert areas (*Witt, Threloff & Hebert, 2006*; *Seidel, Lang & Berg, 2009*); ancient lakes (*Wysocka et al., 2013*; *Grabowski, Wysocka & Mamos, 2017*); and subterranean waters (*Trontelj et al., 2009*), but also in numerous river systems throughout Europe (*Weiss et al., 2014*; *Mamos et al., 2016*). However, most of the above papers did not aim to explain the evolutionary and geological history behind the observed diversity. Such studies are still relatively scarce and usually limited to higher taxonomic levels (*Hou et al., 2011*; *Hou, Sket & Li, 2013*; *McInerney et al., 2014*; *Wysocka et al., 2014*). Only very recently, the onset of diversification in two widespread freshwater morphospecies, *Gammarus balcanicus* Schäferna, 1922 and *Gammarus fossarum* Koch, 1836, in Central and Southern Europe was studied and interpreted in the context of regional geological history (*Copilaş-Ciocianu & Petrusek, 2015*; *Copilaş-Ciocianu & Petrusek, 2017*; *Mamos et al., 2016*). All these studies have stressed that the diversification of both species is very old

and dates back to early/middle Miocene (20–15 Ma), when the continentalisation of the area took place due to the Paratethys regression.

*Gammarus roeselii* Gervais, 1835 is an epigean morphospecies, characterised by rather high morphological polymorphism, commonly found in lakes and rivers of the western Balkan Peninsula (*Karaman & Pinkster, 1977a*; own unpublished data). Only in historical times, the species has expanded its range to the rivers of Western and Central Europe, from France and Netherlands in the west, to Poland in the east (*Grabowski, Jazdzewski & Konopacka, 2007* and references therein). The species is characterised by relatively wide tolerance to various environmental factors and some resistance to pollution (*Gergs, Schlag & Rothhaupt, 2013*). It is also easy to tell apart from other local morphospecies of *Gammarus* by the presence of large dorsal spines on the metasome part of the body, combined with rich setation of appendages (*Karaman & Pinkster, 1977a*). The morphospecies is also well defined in phylogenetic terms; being monophyletic relative to other species of *Gammarus* in Europe (*Hou et al., 2011*). The well-defined distribution of *G. roeselii* in a region with long and dynamic geology makes it a suitable model species for studying the role of past geological events in shaping its present molecular diversity.

In the present paper we aim to use the inferred paleogeographic distribution of ancestors as a framework for understanding the onset of diversification in *Gammarus roeselii*. First, we hypothesise that, taking into account the high number of isolated waterbodies on the Balkans, the species will be characterised by high levels of cryptic diversity; even at a relatively small geographic scale. Second, given the long and turbulent geological history of the region, we hypothesise that most of the diversification and speciation events precedes the Pleistocene Ice Ages. Third, provided *G. roeselii* is able to thrive in both lakes and rivers, we hypothesise that its evolutionary history may be linked to the Balkan Neogene paleolake system and its further deterioration during the geological history of the region. Finally, we inspect whether the Pleistocene decline of hydrological networks in the Balkans could have any impact on the diversification of *G. roeselii*.

## MATERIAL AND METHODS

### Sampling and taxonomic identification

A total of 348 sites located in springs, streams, rivers and lakes all over the Balkan Peninsula, including the Peloponnese (Fig. 1) were surveyed in the years 2006–2008, using semi-quantitative effort, based on a kick sampling method with a benthic hand-net (e.g., *Jazdzewski, Konopacka & Grabowski, 2004*). All the collected material was sorted on-site and gammarids were immediately fixed in 96% ethanol. In the laboratory, animals were identified to species level using morphological characters described in available keys (e.g., *Karaman & Pinkster, 1977a*; *Karaman & Pinkster, 1977b*; *Karaman & Pinkster, 1987*; *Pinkster, 1993*). *Gammarus roeselii* was found at 26 sites (Table 1). All the material used in this study has been stored in the permanent collection of the Department of Invertebrate Zoology and Hydrobiology, University of Lodz, Poland.

Grabowski et al. (2017), *PeerJ*, DOI 10.7717/peerj.3016

**Table 1  Sampling sites features and distribution of haplotypic diversity.** COI = mtDNA cytochrome oxidase I and 28S = nuclear 28S ribosomal DNA. N, sampling size for each marker. MOTU, each haplotype is ascribed to one of the 13 identified MOTUs (A–M). Hap-Acc num(n), haplotype name—Genbank Accession number (number of individuals).

| No | Country | Site | River basin | Sea basin | Coordinates (N, E) | Altitude (m) | COI | | | 28S[a] | |
|----|---------|------|-------------|-----------|---------------------|--------------|-----|-----|-----|-----|-----|
| | | | | | | | N | MOTU | Hap-Acc num(n) | N | Hap-Acc num(n) |
| 1 | SL | Čreta, tributary of Drava River | Danube | Black | 46.544167, 15.614444 | 360 | 3 | C | **C28-**KP789695(3)[b] | – | |
| 2 | SL | Nova Vas, Drava River | Danube | Black | 46.382111, 15.939428 | 215 | 2 | C | C25-KP789694(1) | – | |
| | | | | | | | | C | **C28-**KP789695(1) | | |
| 3 | CR | Varaždin, tributary of Drava River | Danube | Black | 46.319815, 16.359331 | 170 | 1 | C | C23-KP789692(1) | – | |
| 4 | RO | Makovişte, Vicinic River | Danube | Black | 44.942533, 21.661000 | 173 | 7 | C | C24-KP789693(7) | 2 | N6-KP789752(1) |
| | | | | | | | | | | | N7-KP789753(1) |
| 5 | SE | Ilince near Preševo, tributary of Binačka Morava River | Danube | Black | 42.354444, 21.595278 | 750 | 1 | B | C52-KP789715(1) | 1 | N23-KP789769(1) |
| 6 | AL | Shkoder, Drin River | Drin | Adriatic | 42.024833, 19.519133 | 2 | 6 | GG | **C32-**KP789697(1) | – | |
| | | | | | | | | G | C44-KP789708(3) | | |
| | | | | | | | | | C45-KP789709(2) | | |
| 7 | AL | Lin, Ohrid Lake | Drin | Adriatic | 41.068617, 20.645483 | 682 | 5 | G | C31-KP789696(1) | 1 | **N3-**KP789749(1) |
| | | | | | | | | G | **C32-**KP789697(4) | | |
| 8 | AL | Gollomboç, Prespa Lake | Isolated/Drin[c] | Adriatic | 40.861083, 20.940450 | 178 | 4 | G | C34-KP789698(1) | – | |
| | | | | | | | | G | **C35-**KP789699(1) | | |
| | | | | | | | | G | C40-KP789704(1) | | |
| | | | | | | | | G | C41-KP789705(1) | | |

Grabowski et al. (2017), *PeerJ*, DOI 10.7717/peerj.3016

**Table 1** (*continued*)

| No | Country | Site | River basin | Sea basin | Coordinates (N, E) | Altitude (m) | COI | | | 28S[a] | |
|----|---------|------|-------------|-----------|---------------------|--------------|-----|-----|-----|--------|-----|
| | | | | | | | N | MOTU | Hap-Acc num(n) | N | Hap-Acc num(n) |
| 9 | MA | Oteševo, Prespa Lake | Isolated/Drin[c] | Adriatic | 40.969183, 20.912217 | 843 | 10 | G | **C35**-KP789699(3) | 2 | **N3**-KP789749(1) N9-KP789755(1) |
| | | | | | | | | G | **C38**-KP789702(6) | | |
| | | | | | | | | G | **C42**-KP789706(1) | | |
| 10 | GR | Microlimni, Micri Prespa Lake | Isolated/Drin[c] | Adriatic | 40.745067, 21.114217 | 864 | 3 | G | **C38**-KP789702(1) | – | |
| | | | | | | | | G | C39-KP789703(1) | | |
| | | | | | | | | G | **C42**-KP789706(1) | | |
| 11 | AL | Përrenjas, tributary of Shkumbin River | Shkumbin | Adriatic | 41.074583, 20.487933 | 463 | 4 | E | C22-KP789691(1) | 1 | N2-KP789748(1) |
| | | | | | | | | G | C36-KP789700(1) | | |
| | | | | | | | | G | C37-KP789701(1) | | |
| | | | | | | | | G | C43-KP789707(1) | | |
| 12 | AL | Orikum, Dukatit River | Dukatit | Adriatic | 40.331967, 19.477667 | 9 | 4 | H | C59-KP789720(1) | 4 | N1-KP789747(4) |
| | | | | | | | | H | C60-KP789721(1) | | |
| | | | | | | | | H | C61-KP789722(1) | | |
| | | | | | | | | H | C62-KP789723(1) | | |
| 13 | AL | Zvezdë, Devoll River | Seman | Adriatic | 40.708033, 20.871200 | 828 | 12 | E | C48-KP789711(1) | 10 | N4-KP789750(8) N5-KP789751(2) |
| | | | | | | | | E | C49-KP789712(4) | | |
| | | | | | | | | E | C50-KP789713(1) | | |
| | | | | | | | | E | C51-KP789714(2) | | |
| | | | | | | | | J | C63-KP789724(1) | | |
| | | | | | | | | J | C64-KP789725(3) | | |

Grabowski et al. (2017), *PeerJ*, DOI 10.7717/peerj.3016

**Table 1** (*continued*)

| No | Country | Site | River basin | Sea basin | Coordinates (N, E) | Altitude (m) | COI | | | 28S[a] | |
|----|---------|------|-------------|-----------|--------------------|--------------|-----|---|---|--------|---|
| | | | | | | | N | MOTU | Hap-Acc num(n) | N | Hap-Acc num(n) |
| 14 | GR | Soulopoulo, tributary of Kalamas (Thyamis) river | Kalamas (Thyamis) | Ionian | 39.718800, 20.610083 | 167 | 10 | M | C84-KP789740(1) | 3 | N11-KP789757(1) N12-KP789758(1) |
| | | | | | | | | M | C85-KP789741(1) | | |
| | | | | | | | | M | C86-KP789742(2) | | |
| | | | | | | | | M | C87-KP789743(1) | | |
| | | | | | | | | M | C88-KP789744(1) | | |
| | | | | | | | | M | C89-KP789745(1) | | |
| | | | | | | | | M | C90-KP789746(3) | | |
| 15 | GR | Platanias, Trichonida (Trichonis) Lake | Acheloos | Ionian | 38.596283, 21.537317 | 5 | 12 | I | C65-KP789726(1) | 5 | N13-KP789759(2) |
| | | | | | | | | I | C66-KP789727(1) | | N14-KP789760(2) |
| | | | | | | | | I | C67-KP789728(1) | | N15-KP789761(1) |
| | | | | | | | | I | C68-KP789729(1) | | |
| | | | | | | | | I | C69-KP789730(1) | | |
| | | | | | | | | I | C70-KP789731(7) | | |
| 16 | MA | Canion Matka, Treska River | Vardar (Axios) | Aegean | 41.963483, 21.301367 | 279 | 9 | A | C7-KP789679(7); | 3 | N8-KP789754(3) |
| | | | | | | | | D | C21-KP789690(2) | | |
| 17 | GR | Petres, Petron Lake | Aliakmonas | Aegean | 40.728267, 21.681150 | 585 | 9 | A | C1-KP789673(6) | – | |
| | | | | | | | | A | C2-KP789674(2) | | |
| | | | | | | | | A | C9-KP789680(1) | | |
| 18 | GR | Agios Panteleimon, Vegoritis Lake | Aliakmonas | Aegean | 40.738600, 21.754600 | 290 | 9 | A | C3-KP789675(1) | 1 | N10-KP789756(1) |
| | | | | | | | | A | C4-KP789676(4) | | |
| | | | | | | | | A | C17-KP789687(1) | | |
| | | | | | | | | A | C18-KP789688(2) | | |
| | | | | | | | | A | C19-KP789689(1) | | |
| 19 | BU | Drangovo-Marikostinovo, Struma (Strymonas) River | Struma (Strymonas) | Aegean | 41.413517, 23.320050 | 103 | 11 | A | C10-KP789681(1) | – | |

**Table 1** (*continued*)

| No | Country | Site | River basin | Sea basin | Coordinates (N, E) | Altitude (m) | COI | | | 28S[a] | |
|---|---|---|---|---|---|---|---|---|---|---|---|
| | | | | | | | N | MOTU | Hap-Acc num(n) | N | Hap-Acc num(n) |
| | | | | | | | | A | C11-KP789682(1) | | |
| | | | | | | | | A | C12-KP789683(1) | | |
| | | | | | | | | A | **C13-KP789684**(7) | | |
| | | | | | | | | A | C16-KP789686(1) | | |
| 20 | GR | Petritsio, Struma (Strymonas) River | Struma (Strymonas) | Aegean | 41.281817, 23.332333 | 60 | 7 | A | **C13-KP789684**(2) | 3 | N22-KP789768(3) |
| | | | | | | | | A | C15-KP789685(1) | | |
| | | | | | | | | E | C46-KP789710(4) | | |
| 21 | GR | Kastoria, Kastoria (Orestiada) Lake | Aliakmonas | Aegean | 40.514227, 21.268950 | 510 | 5 | A | C5-KP789677(1) | – | |
| | | | | | | | | A | **C6-KP789678**(4) | | |
| 22 | GR | Near Siatisti, Aliakmonas River | Aliakmonas | Aegean | 40.288867, 21.451133 | 553 | 5 | A | **C6-KP789678**(5) | – | |
| 23 | GR | Aleksandria, Aliakmonas River | Aliakmonas | Aegean | 40.583550, 22.466033 | 17 | 11 | F | C53-KP789716(1) | 2 | N21-KP789767(2) |
| | | | | | | | | F | C54-KP789717(1) | | |
| | | | | | | | | F | C55-KP789718(1) | | |
| | | | | | | | | F | C56-KP789719(8) | | |
| 24 | GR | Paliouri, Sperchios River | Sperchios | Aegean | 38.943450, 22.211767 | 36 | 9 | K | C74-KP789734(4) | 2 | N16-KP789762(1) |
| | | | | | | | | K | C76-KP789735(5) | | **N17-KP789763**(1) |
| 25 | GR | Omolio, Pinios River | Pinios | Aegean | 39.878333, 22.584550 | 35 | 9 | L | C78-KP789736(1) | 3 | N19-KP789765(2) |
| | | | | | | | | L | C79-KP789737(1) | | N20-KP789766(1) |
| | | | | | | | | L | C80-KP789738(5) | | |
| | | | | | | | | L | C83-KP789739(2) | | |
| 26 | GR | Kedros, Sofaditikos River | Pinios | Aegean | 39.176200, 22.045983 | 435 | 9 | K | C72-KP789732(8) | 8 | **N17-KP789763**(7) |
| | | | | | | | | K | C73-KP789733(1) | | N18-KP789764(1) |

**Notes.**

Country code: SL, Slovenia; CR, Croatia; RO, Romania; SE, Serbia; AL, Albania; MA, Macedonia; BU, Bulgaria; GR, Greece.

[a]Note that sampling for 28S was according to the MOTUs identified by COI, not geography.

[b]In bold—haplotypes shared by two sites.

[c]No connection via surface waters, but connected to the Ohrid Lake via karst underground channel.

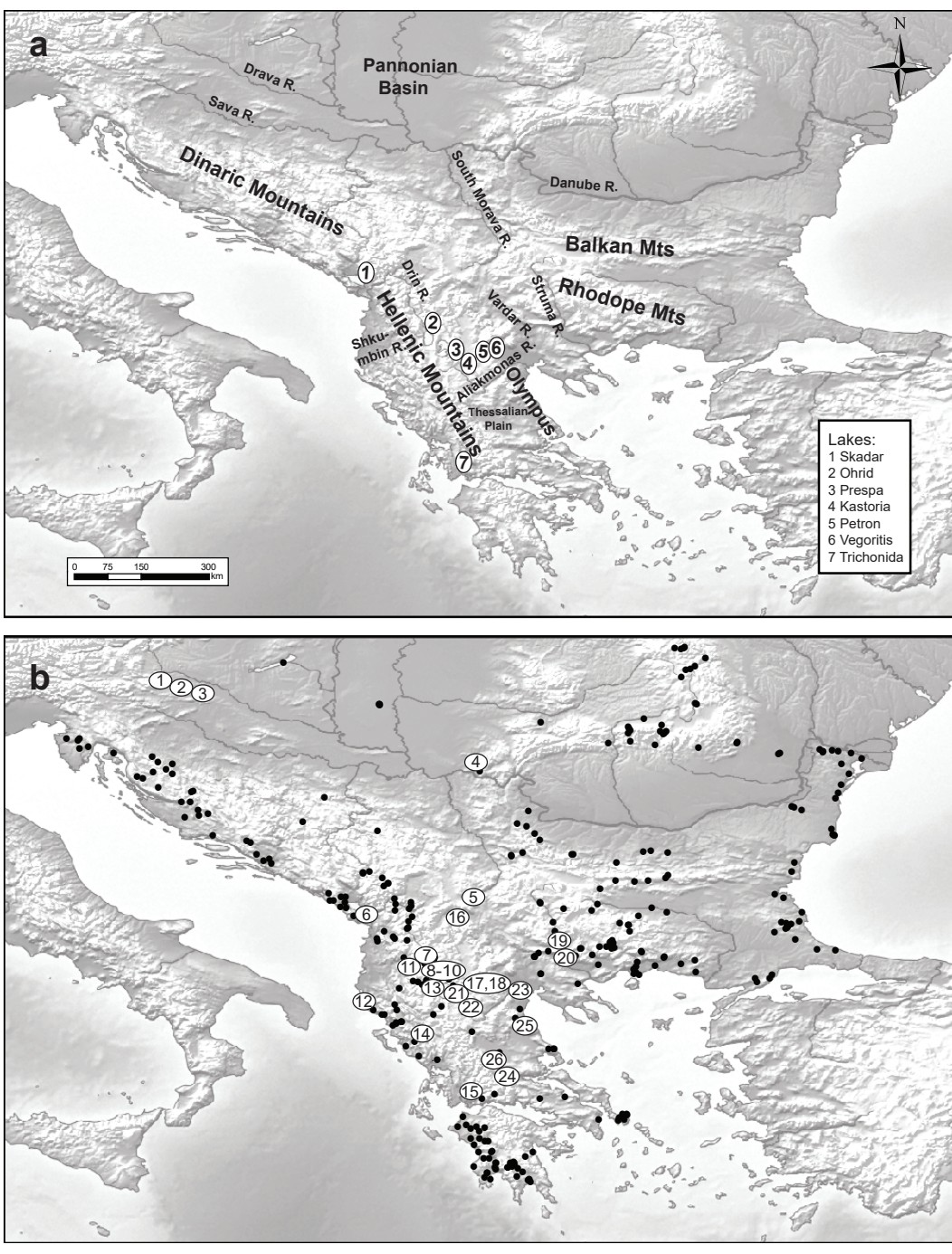

**Figure 1  Maps of the studied area and sampling sites.** Maps of (A) the studied area and (B) the 348 sampled sites (black dots denote the absence of *Gammarus roeselii* and sites numbered 1–26 the sites where the species was present.

## Molecular procedures (DNA extraction, PCR amplification and sequencing)

Genomic DNA was extracted from pieces of muscle tissue obtained from 1 to 12 individuals from each site, making up a total of 177 individuals (Table 1). All individuals were processed, via PCR, to amplify a ca. 650 bp fragment of the mtDNA cytochrome C oxidase subunit I (COI), using the primer pairs: LCO1490 and HCO2198 (*Folmer et al., 1994*); UCOIF and UCOIR (*Costa et al., 2009*); and a newly designed primer pair COIGrF (5′-GCTAGH GCCGTAGGYACATC) and COIGrR2 (5′-RAATARGTGYTGGTACAGAATAGG). Details about which primers were specifically used for each individual are provided in Table S1. After defining molecular operational taxonomic units (MOTU) based on the COI marker (see 'Results'), a subset of 50 individuals representing all the defined MOTUs was also amplified to provide a ca. 900 bp long fragment of the 28S nrDNA using 28F and 28R primers (*Hou, Fu & Li, 2007*). An aliquot of each PCR product was used to check the specificity and yield by agarose gel electrophoresis. PCR products were EXO-FastAP (Thermo Scientific) purified prior to sequencing by Macrogen Inc., Korea, using BigDye terminator technology. All the details of the molecular procedures were as described in *Mamos et al. (2016)*.

## Sequence data authentication, editing and alignment

All sequences were positively verified as *Gammarus* DNA via BLASTn searches against GenBank (*Altschul et al., 1990*). Geneious 6.1.4 (*Biomatters Inc, 2013*) was used for sequence editing, trimmed to either 530 nucleotides (COI) or 563 nucleotides (28S) and aligned using Geneious alignment algorithm. Haplotypes were identified for each marker with the DnaSP software (*Librado & Rozas, 2009*). All single marker haplotypic sequences were deposited in GenBank (Accession numbers in Table 1). For COI, Kimura 2-parameter (K2p) distance was calculated (e.g., between all haplotypes, or within and among MOTUs) using MEGA 6.0 (*Tamura et al., 2013*). Standard error estimates were obtained by a bootstrap procedure (1,000 replicates).

## Time-calibrated reconstruction of phylogeny

The time-calibrated phylogeny was reconstructed using the Bayesian inference (BI) in BEAST 1.8.2 (*Drummond et al., 2012*). Two amphipod species *Dikerogammarus villosus* (Sovinsky, 1894) (EF570297) and *Pontogammarus robustoides* (GO Sars, 1894) (JF965990) were used as out-groups for *G. roeselii* in the following analysis. Given that no molecular clock calibration based on gammarid fossil material is available, the calibration was performed applying three different schemes in parallel: (1) secondary calibration points; (2) standard rate of evolution; and (3) geological calibration points. The first scheme was based on the secondary calibration points from *Hou et al. (2011)*. We constrained the node between *Gammarus* sp. 3 (JF965959) and *Gammarus rambouseki* Karaman 1931 (JF965946) to 15.5 Ma. The node between these two species and our sequences of *G. roeselii* was constrained to 22 Ma. Dating constraints were set to normal distributions with 10% standard deviation. The second calibration scheme applied the "standard" rate of the COI evolution as 0.0115 per site per My (*Brower, 1994*). The third scheme was based on two

geological calibration points: (a) withdrawal of Moesian Sea and subsequent formation of modern Danube river system; (b) origins of ancient Lake Baikal. The calibration points, as well as the prior settings, were described in detail in *Mamos et al. (2016)*. For the first calibrated point (a) we used sequences of *G. cf. balcanicus* from the Southern Carpathians (KU056394) and the Balkan Mountains (KU056256). The constraint was set to the time between the withdrawal of the Moesian Sea at ca. 2.6 Ma and emergence of the modern Danube system at ca. 1.3 Ma, which connected formerly-isolated rivers of the Southern Carpathians and the Balkan Mountains (*Miklos & Neppel, 2010*; *Mamos et al., 2016*). For the second calibration point (b) we used the common ancestor of Acanthogammarids from the ancient Lake Baikal. A permanent lake in the Baikal basin that originated probably ca. 27 Ma, but the conditions in the lake were subsequently changing. Thus, the lacustrine species flocks may be much younger than the lake itself (*Mats et al., 2000*). According to *Mats, Shecherbakov & Efimova (2011)*, the radiation of Acanthogammarids, represented here by *Acanthogammarus brevispinus* Dorogostaisky, 1922 (AY926651), *Ommatogammarus albinus* Dybowsky, 1874 (AY926686) and *Eulimnogammarus viridulus* Bazikalova, 1945 (AY926665), started at c. 17 Ma. Thus, the constraints were designed to produce results centred on this value but ranging back to the origins of the lake.

In all three calibration schemes the strict clock (vs relaxed lognormal and relaxed exponential), general time-reversible (GTR) model of evolution with gamma-distributed rate heterogeneity (G) and a proportion of invariable sites (I) model of evolution (vs all other models available in BEAST) as well as the Birth-Death speciation process (vs Yule) were set as priors, following the Akaike information criterion (AIC) through the Markov chain Monte Carlo (MCMC) method of moments estimator (*Baele et al., 2013*) in Tracer 1.6 (*Rambaut et al., 2014*). Four MCMC, each with 20 M iterations, were run and sampled every 2,000 iterations. The Effective Sampling Size (ESS) of each parameter was verified to be above 200 in Tracer 1.6 (*Rambaut et al., 2014*). Runs were combined with LogCombiner 1.8.2 (*Drummond et al., 2012*) with 25% burn-in, and the maximum clade credibility chronogram was annotated using TreeAnnotator 1.8.2 (*Drummond et al., 2012*). Since all the time-calibrated phylogeny reconstructions provided congruent results (Table S2) we have used the tree obtained from the first calibration scheme as a proxy for following analysis of MOTU delimitation and Bayesian reconstruction of ancestral states. Our interpretation of the obtained results in the context of local paleogeography follows the established and commonly accepted intercalibrations of direct and stratigraphic dating provided by *Gradstein, Ogg & Smith (2004)* and by *Piller, Harzhauser & Mandic (2007)*, which were summarised for the area by *Harzhauser & Mandic (2008)*.

In order to provide additional support for the BI topology we have also reconstructed a phylogeny using the Maximum Likelihood approach (ML) in MEGA6 software (*Tamura et al., 2013*). The GTR + G + I model of evolution was used with 1,000 bootstrap replicates (*Felsenstein, 1985*).

## Analysis of diversification rates

The history of diversification was explored using the lineage through time (LTT) plot generated from 1000 BI trees in Tracer 1.6. In order to avoid artificial influence of

coalescence on terminal part of the plot we have used BI trees generated with the *BEAST algorithm (*Heled & Drummond, 2010*) in BEAST 1.8.2. Species delimited for the analysis were coded according to the consensus MOTUs. The BI analysis was conducted using the same priors and MCMC settings as in the first calibration scheme of the time calibrated reconstruction of phylogeny. To identify the potential shifts in diversification we used Laser 2.3 package (*Rabosky, 2006*) written in R (*R Core Team, 2012*), a set of maximum likelihood-based methods for analyzing lineage diversification rates. We estimated the fit of six speciation models (function fitdAICrc), assuming constant (pure birth, birth death) and variable (Yule 2-rate, Yule 3-rate, exponential and logistic density dependent) diversification rates, to the reconstructed BI phylogeny.

## Cryptic diversity—MOTU delimitation

To explore the number of MOTUs that could represent putative cryptic species within the morphospecies *G. roeselii*, we applied five different methods: (i) a distance based Barcode Index Number (BIN) System (*Ratnasingham & Hebert, 2013*); (ii) a distance-based barcode-gap approach using the Automatic Barcode Gap Discovery (ABGD) software (*Puillandre et al., 2012*); (iii) the tree-based, phylogenetic approach using the Bayesian implementation of the Poison Tree Processor (bPTP) (*Zhang et al., 2013*) and (iv and v) the tree-based general mixed Yule coalescent (GMYC) model-based method (*Pons et al., 2006*), using either single and multiple threshold models.

The BIN method is implemented as part of The Barcode of Life Data systems (BOLD; *Ratnasingham & Hebert, 2007*). Newly submitted sequences are compared together as well as with sequences already available in BOLD. Sequences are clustered according to their molecular divergence using algorithms aiming at finding discontinuities between clusters. Each cluster is ascribed a globally unique and specific identifier (aka Barcode Index Number or BIN), already available or newly created if newly submitted sequences do not cluster with already known BINs. Each BIN is registered in BOLD.

The ABGD method is based upon pairwise distance measures. With this method the sequences are partitioned into groups (MOTUs), such that the distance between two sequences from two different groups will always be larger than a given threshold distance (i.e., barcode gap). We used primary partitions as a principal for group definition, as they are typically stable on a wider range of prior values, minimise the number of false positive (over-split species) and are usually close to the number of taxa described by taxonomists (*Puillandre et al., 2012*). The default value of 0.001 was used as the minimum intraspecific distance. As there is currently no consensus about which maximum intraspecific distance is reflecting delimitation of species, neither based on morphology (*Costa et al., 2007*; *Weiss et al., 2014*, or *Katouzian et al., 2016*) nor on reproductive barrier (*Lagrue et al., 2014*) we explored a set of values up to 0.1. The standard Kimura two-parameter (K2p) model correction was applied (*Hebert et al., 2003*).

The tree based bPTP method for species delimitation uses non-ultrametric phylogenies. The bPTP method incorporates the number of substitutions in the model of speciation and assumes that the probability that a substitution gives rise to a speciation event follows a Poisson distribution. The branch lengths of the input tree are supposed to be generated

by two independent classes of the Poisson events, one corresponding to speciation and the other to coalescence. Additionally the bPTP adds Bayesian support values (BS) for the delimited species (*Zhang et al., 2013*). For the input tree we generated phylogeny using Bayesian inference in MrBayes 3.2 (*Ronquist et al., 2012*). Prior to tree construction, the out-group was removed from the alignment. The GTR + G + I model was selected using AICM method of moments estimator (*Baele et al., 2013*) in Tracer 1.6 (*Rambaut et al., 2014*). Four MCMC, each with 20 M iterations, were run and sampled every 2,000 iterations. This was enough to obtain potential scale reduction factor values equal 1 for all parameters. Stable convergence was determined by inspecting the log likelihood values of the chains and the split frequencies that reach, after 10 million generations, value above 0.01. The consensus tree was constructed after removal of 25% burn-in phase. Analysis was performed on the bPTP web server (available at: http://www.species.h-its.org/ptp/) with 500,000 iterations of MCMC and 10% burn-in.

With the GMYC method, species boundaries are assessed based on sequences in a maximum likelihood framework by identifying the switch from intraspecific branching patterns (coalescent) to typical species branching patterns (Yule process) on a phylogenetic tree. First, a log-likelihood ratio test is performed to assess if the GMYC mixed models fit the observed data significantly better than the null model of a single coalescent species. Providing there is evidence for overlooked species inside the phylogenetic tree tested, two different GMYC approaches, one using the single threshold and the other one on multiple threshold model are used to estimate the boundary between intra- and interspecific branching patterns. Since the GMYC-method requires an ultrametric tree, we have used the already reconstructed Bayesian time calibrated phylogeny. The consensus tree was loaded into the R software package 'SPLITS' (Species Limits by Threshold Statistics) (*Ezard, Fujisawa & Barraclough, 2009*) in R v3.1.0 (*R Core Team, 2012*) and analysed using the single and multiple threshold models. Presence of significant differences between the two models was tested using likelihood ratio test (LRT).

## Bayesian reconstruction of the ancestral states

The program BayesTraits (*Pagel, Meade & Barker, 2004*) was used to implement the MCMC Bayesian method, aiming at reconstructing ancestral states of a given trait along a phylogenetic tree. Two traits, habitat (river *vs* lake) and geographic coordinates (longitude and latitude) were explored. The BI maximum clade credibility chronogram was used as the input tree and the actual habitat type and geographic coordinates of its tips were used as input data for the actual state of the trait. For each ancestral node along the tree the results were reported as (i) a chance (in percentage) for the ancestor to occur in a given habitat or (ii) possible coordinates of the ancestor's locality. In both cases the analysis was performed in two steps. Firstly, using the mentioned input data, a distribution of models was estimated and saved into a file. Secondly, the input data with file containing models distribution was used in the final analysis in which four independent, 10 M iterations long, MCMC runs were performed and sampled every 1,000 iterations, with 5% burn-in. Continuous and multistate reconstruction options with equal priors models, were used for geographic coordinates and habitat, respectively. All runs in each analysis produced convergent results

that were combined and summarised (see Table S3 in Supporting Information). Mean and standard deviations for the ancestral geographic coordinates were calculated in MS Excel.

### Phylogenetic tree of 28S haplotype and relationship with COI MOTUs

To display the relationships between the nuclear 28S rDNA haplotypes a phylogenetic Neighbor-Joining (NJ) tree based p distance and node robustness being evaluated by 1,000 bootstraps was constructed using MEGA 6.0 (*Tamura et al., 2013*). To display relationship (i.e., discrepancy between phylogenies) between nuclear and mitochondrial phylogenies COI MOTUs were plotted on 28S NJ tree.

### Within MOTU diversity, divergence, differentiation and historical demography: a Pleistocene tale

Molecular genetic diversity, divergence, differentiation and historical demography based on mtDNA COI sequences were, wherever possible, estimated for each MOTU. The relative frequency, geographic distribution and molecular divergence between COI haplotypes, were graphically presented as a median-joining (MJ) network (*Bandelt, Forster & Rohl, 1999*) implemented in the program NETWORK 4.6.1.2 (http://www.fluxusengineering.com). Given the differences in substitution rates, we applied a weight of 1 for transition and 2 for transversions. The topology was obtained at the homoplasy level parameter default value ($e = 0$).

Diversity was assessed as the number of haplotypes ($k$), haplotypic diversity ($h$) and nucleotide diversity ($\pi$) (*Nei, 1987*) with the DnaSP software (*Librado & Rozas, 2009*). Molecular divergence was estimated as average Kimura 2—parameters (K2p) distance between haplotypes using MEGA 6.0 (*Tamura et al., 2013*). In case of MOTUs present in more than one site (MOTUs A, C, E, G and K) and given a minimum sampling size per site of 3 individuals, the differentiation in haplotype frequencies was estimated as overall $F_{ST}$ through an Analysis of Molecular Variance (AMOVA, *Excoffier, Smouse & Quattro, 1992*) with 10,000 permutations. In addition, pairwise differentiation between locations was estimated through $F_{ST}$ (*Weir & Cockerham, 1984*) with 1,023 permutations, and through Exact Tests of Sample Differentiation (ETSD, *Raymond & Rousset, 1995*) with 30,000 Markov steps. Historical demographic expansion within MOTUs at both the scale of MOTUs and location was examined based on Tajima's $D$ (*Tajima, 1989*) and Fu's $Fs$ (*Fu, 1996*) neutrality tests with 1,000 replicates. All the above analyses were performed in Arlequin 3.5.1.3 (*Excoffier & Lischer, 2010*).

## RESULTS

### Species geographic distribution and associated habitat

We collected *Gammarus roeselii* from 26 sites on the Balkan Peninsula, including 18 riverine and 8 lacustrine sites belonging to 12 river basins draining to the Black, Adriatic, Ionian and Aegean seas as well as the isolated Prespa lake (Table 1, Fig. 1). The altitudinal range of the sites spans from the sea level to nearly 870 m.a.s.l. The longitudinal range of sites in which *G. roeselii* was found extended from the Adriatic/Ionian coast of the Balkan Peninsula (15.7°E) on the west to the Struma river system (24°E), and the latitudinal range extended

from the Drava River (46°N) down to the Corinthian Bay (38.5°N). Despite extensive search, *G. roeselii* was not found neither in the eastern part of the Balkan Peninsula nor on the Peloponnese Peninsula. Other areas where we did not encounter presence of *G. roeselii* included central Serbia, Bosnia & Herzegovina and coastal regions of Croatia.

### COI haplotypes: diversity, distribution and divergence

Out of 177 individuals sequenced, a total of 74 haplotypes were identified (Accession numbers KP780673–KP780746) (Table 1). Only seven haplotypes were shared between a pair of locations (Table 1). Each of the remaining 67 haplotypes was specific to only one location (Table 1). Minimum, average and maximum K2p distance between haplotypes were, respectively, 0.0019 (SE = 0.0017), 0.1463 (SE = 0.0108) and 0.2405 (SE = 0.0108). Such high average K2p distance suggests presence of cryptic diversity within the morphospecies (see also Table S4).

### Time calibrated reconstruction of phylogeny and analysis of diversification rate

To infer the chronology of evolutionary events of diversification in *G. roeselii*, we performed a Bayesian time-calibrated reconstruction of phylogeny based on the COI marker. Both the Bayesian inferred (BI) chronogram and the Maximum Likelihood (ML) tree constructed in parallel had merely the same topology and high node supports (Fig. 2). The mean substitution rate associated with BI chronogram was 0.0113 in substitutions per site, per My, (SD 0.0016, 95% HPD 0.0083–0.0144) for first calibration scheme and 0.0129 substitutions per site, per My (SD 0.0030, 95% HPD 0.0077–0.0193) for third calibration scheme (Table S2).

The spatio-temporal pattern of lineage diversification within *G. roeselii* in the Balkan Peninsula appeared to be very complex (Fig. 2A, Table S1). The diversification started ca. 18 Ma and produced a high number (>10) of extant lineages. Spatial distribution of these lineages is heterogeneous, most of them are local endemics, two are relatively widespread and few occur in sympatry (Fig. 2B). The initial divergence (ca. 18 Ma) produced two groups of lineages—one spread over the northern part of the Balkan Peninsula, while the other one colonised southern part of this region. Each of the two groups is composed of a set of lineages that diverged before the beginning of Pleistocene glaciations, namely in Miocene (ca. 20.0–5.3 Ma) and Pliocene (ca. 5.3–2.6 Ma). The lineages through time plot (LTT, Fig. 3) shows that the accumulation of lineages was rather constant until Pleistocene. The analysis of diversification rate change through time showed that constant rate pure birth model and density-dependent speciation rate model with logistic variants fit the data best, with only a little difference between them (dAIC = 0.21).

### Cryptic diversity/MOTU definition

The thirteen MOTUs (A–M, Fig. 2) representing a set of potential cryptic species, we retained, are based on the combination of the information brought by (i) five species delimitation methods (BINs, ABGD, bPTP, GMYC single threshold and GMYC multiple threshold model) and (ii) the divergence age on the BI chronogram (Fig. 2).

All the 177 sequences submitted to BOLD clustered into 27 BINs (details in Table S1).

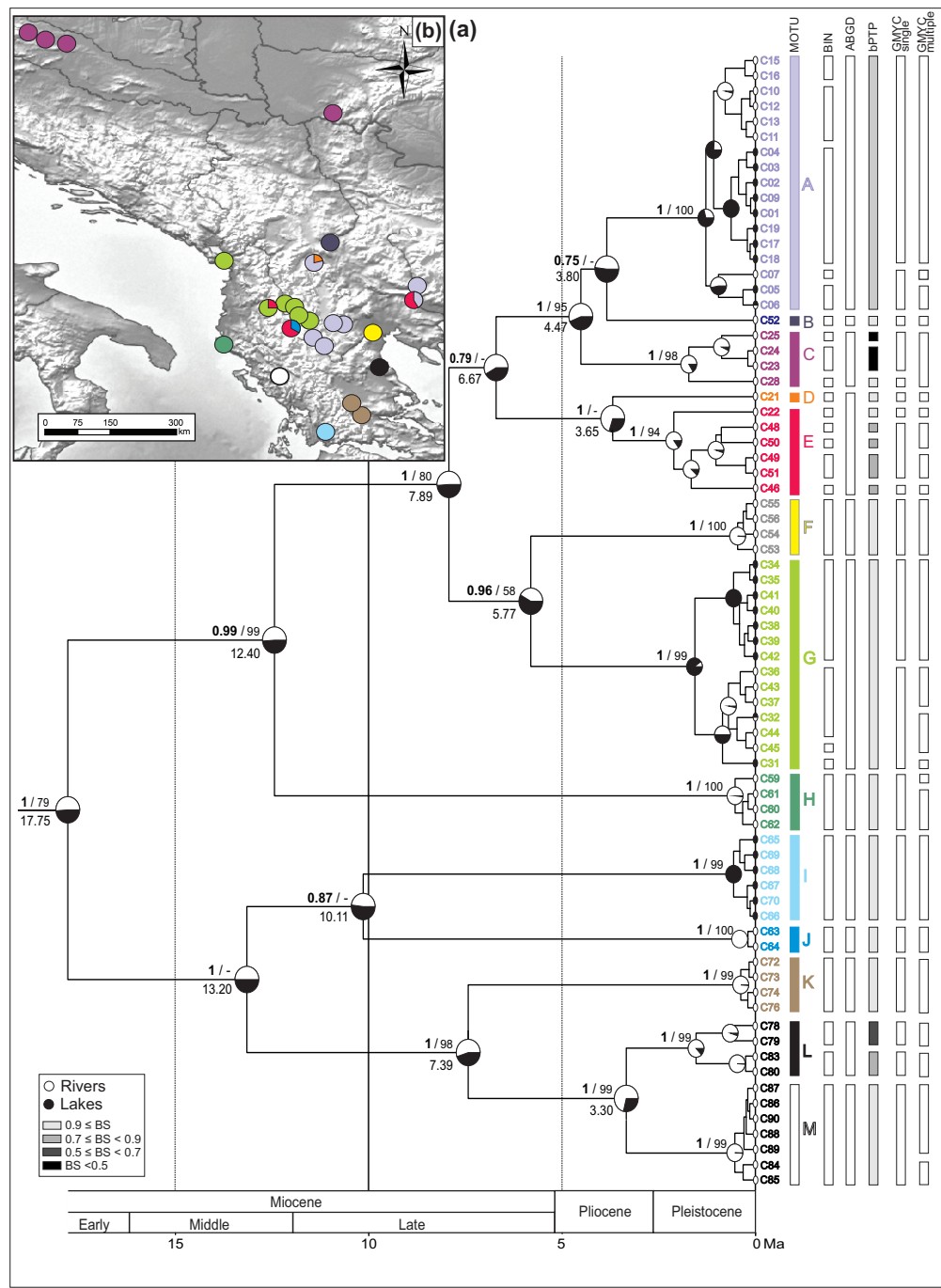

**Figure 2  Bayesian chronogram with MOTU designation.** (A) Bayesian maximum clade credibility chronogram based on COI mtDNA including actual and ancestral state of habitat type (river vs lake), bars annotated on the right represents consensus MOTUs designation and results of the species delimitation methods (BINs, ABGD, bPTP, GMYC-single, GMYC-multiple). Values at nodes above branches are Bayesian posterior probabilities and bootstrap percentage, respectively before and after slash. Values below 50% denoted as –. Values at nodes below branches are split age in Ma. (B) Geographic distribution of the MOTUs.

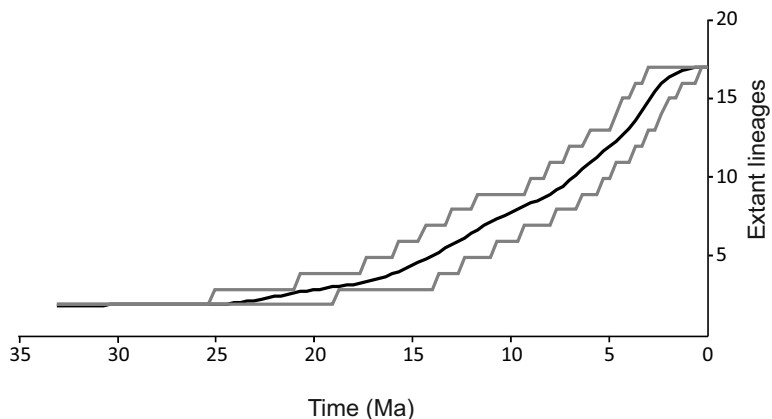

**Figure 3 Lineages through time plot.** Haplotypes were collapsed to consensus MOTUs using *BEAST method. Grey lines represents 95% HPD.

The ABGD distance-based approach (barcode-gap analysis), resulted in eight primary partitions of the analysed *COI* sequences. Four of them were stable over a wide range of *P* values (0.001–0.10) and defined 12 MOTUs, being the same in each partition (Fig. 2).

The estimated number of MOTUs in bPTP ranged between 14 and 29, out of which 20 have Bayesian support values above 0.46. Eight of them are represented only by a single haplotype (Fig. 2).

In the GMYC, the null model of a single species was rejected in both, single and multiple threshold models (GMYC single LRT < 0.002, GMYC multiple LRT < 0.0001). Results of the chi-square test showed that there was no statistically significant difference between the two models (LRT = 6.305, $p = 0.39$). Results of the single threshold analysis suggested the presence of 19 MOTUs (confidence interval: 12–30), out of which five consisted of single haplotypes. The results of the multiple threshold analysis suggested 24 MOTUs (confidence interval: 17–28), out of which seven consisted of single haplotypes (Fig. 2).

The 13 MOTUs retained, corresponded to the MOTUs identified by the ABGD method with the only exception that we decided to split one ABGD MOTU in two MOTUs (D + E). They diverged at ca. 4 Ma, already in the Pliocene, and the K2p distance between these MOTUs is ca. 6%. Such a distance might have been associated with reproductive barrier observed in other *Gammarus* species (*Lagrue et al., 2014*) (Table S4, see Discussion).

## Chronology of divergence and geographical distribution of MOTUs

The identified MOTUs formed two well defined primary clades. The first clade grouped MOTUs (A–H) occupying the northern and the central part of the Balkan Peninsula. The second one included MOTUs (I–M) present predominantly in the southern part of the peninsula. The clades appear to have diverged in the early Miocene (ca. 18 Ma) (Fig. 2A).

Divergence within both clades started ca. 13 Ma, leading to separation of MOTU H from A to G in the northern clade and I–J from K to M in the southern one. Next, within the northern clade, at ca. 8 Ma, the group including MOTUs A–G separated into two subgroups A–E and F–G. The former, at ca. 7 Ma, gave rise to MOTU sets A–C and D–E. The MOTU C separated from the others at ca. 4.5 Ma, while A and B diverged at ca. 4

Ma. MOTUs D and E separated at the same time. Of the other subgroup, MOTUs F and G diverged at ca. 7 Ma. Within the southern clade, MOTUs I and J diverged already at ca. 10 Ma. Then MOTU K separated from L to M at ca. 7 Ma and, finally, MOTU L diverged from M at ca. 3.3 Ma (Fig. 2A).

Currently, for the northern clade, MOTU A had the widest distribution. It was present in seven sites, including rivers and lakes in the central Balkan Peninsula from the Aliakmonas drainage on the southwest, to the Vardar drainage on the north, and the Struma drainage on the east (Fig. 2B). MOTU A coexisted with MOTU D in the Vardar River drainage, the only place where MOTU D was found, and with MOTU E in the Struma River. The lacustrine populations of MOTU A inhabited Kastoria, Vegoritis and Petron lakes; all belonging to the Aliakmonas drainage. MOTU B was found only at one site in the southernmost part of the South Morava drainage, a major Danube tributary (Fig. 2B). MOTU C showed the northernmost distribution, being present only in rivers of the Danube drainage system (Black Sea basin), on the Pannonian Basin. MOTU E was another widely distributed lineage, found in the central part of the Balkan Peninsula. Interestingly, it was always found in sympatry with other MOTUs—with A in the Struma River, with G in the Shkumbin River, and with J in the upper Devoll River (a tributary of the Seman River). MOTU G, present in six sites, was predominantly associated with the Drin River drainage, including also the Ohrid and Prespa lakes, but also the nearby Shkumbin River, draining directly to the Adriatic Sea. MOTU H was found at only one isolated site in a small river on the Adriatic side of the Hellenic Mountains (Fig. 2B).

The MOTUs belonging to the southern group are spatially scattered and have very localised distributions (Fig. 2B). Each of them is restricted usually to only one waterbody or drainage area. For example, the highly divergent MOTU I occupies the southernmost locality. It was found only in the Trichonis Lake, on the Ionian side of the Hellenic Mountains. Its sister MOTU J was present only in a remote locality on the upper Devoll River, in the north-eastern outskirts of the Hellenic Mountains. MOTU K was found in the Sperchios drainage on the Thessalian Plain and in the neighbouring Sofaditikos drainage in the Pelion Mountains. MOTU L was found in the Pinios drainage, also in Thessaly. Interestingly, its sister MOTU M was found only in the remote Thyamis drainage, on the Adriatic side of the Hellenic Mountains.

## Reconstruction of the ancestral longitudes/latitudes and ancestral habitats of MRCAs

Reconstruction of the geographical origin of the most recent common ancestors (MRCAs) (Fig. 4A, Table S3) allowed to project the lineage diversification within *G. roeselii* on the present geography of the Balkan Peninsula and to interpret it in the paleogeographic context. The analysis showed that the early Miocene diversification of *G. roeselii* lineages might have started in the western part of the central Balkan Peninsula. From there, the northern clade could have spread to the neighbouring areas of the present Danube, Drin, Vardar, Shkumbin, Aliakmonas and Struma drainages; where it diversified throughout Miocene and Pliocene. The southern clade had colonised the southwestern parts of the Balkan Peninsula, diversifying in the Miocene and Pliocene, most probably on the

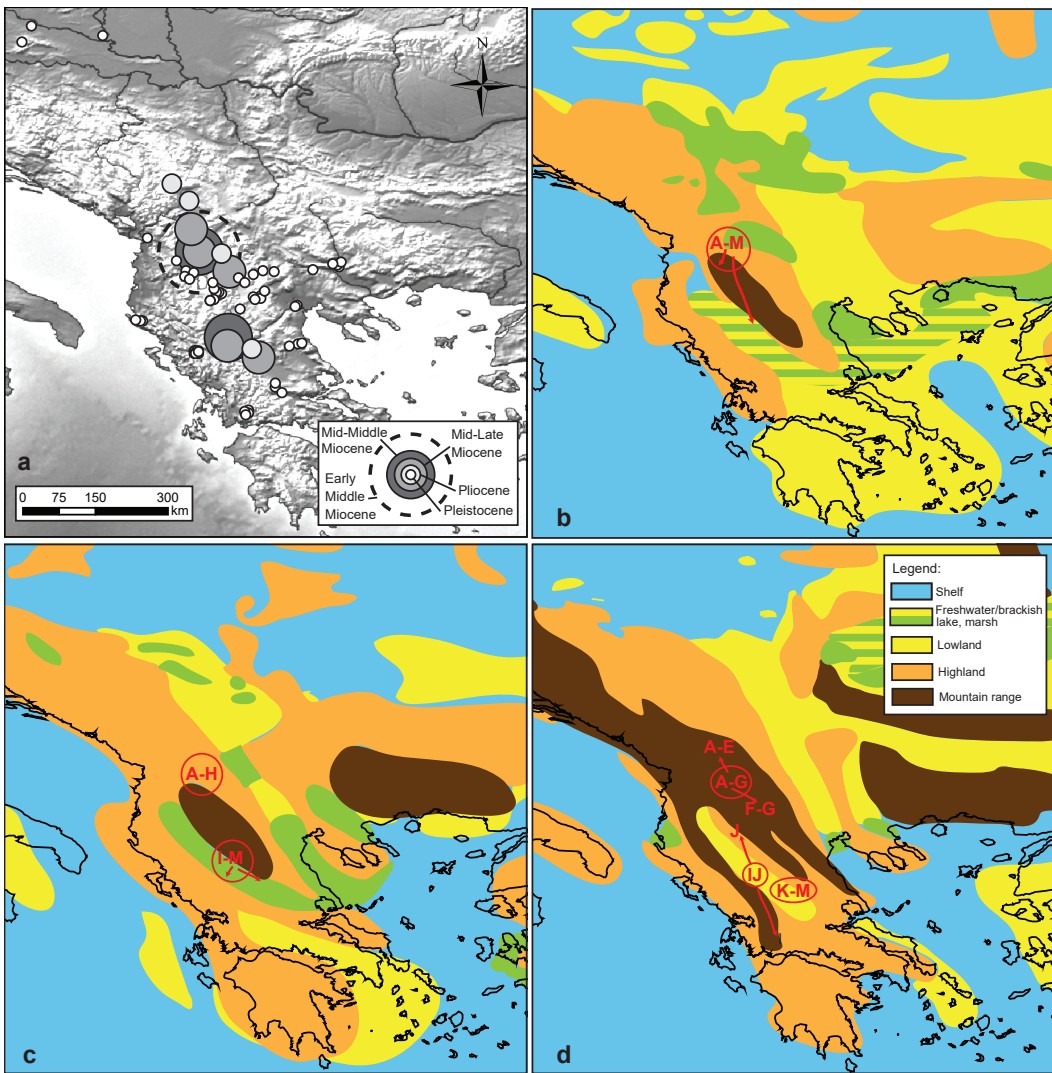

**Figure 4** **Reconstruction of the putative geographic positions of ancestors.** Reconstruction of the putative geographic positions of ancestors based on the COI chronogram: (A) position of the ancestors projected on the present-day geography of the studied area, (B–D) positions of common ancestors (circle) and possible directions of colonization (arrows) projected on palaeomaps (redrawn after *Popov et al., 2004*) showing key geological events: (B) early Middle Miocene 16–15 Ma. (C) middle Middle Miocene 14–13 Ma, (D) mid Late Miocene 7–6 Ma.

eastern side of the Hellenic Mountains, before they effectively crossed the mountain chain and colonised now isolated watersheds. Finally, in the Pleistocene, a number of local diversification events occurred in MOTUs of both clades all over the species distribution on the Balkan Peninsula.

According to the reconstruction of the ancestral habitats of MRCAs (Fig. 2A), most of the Miocene ancestors could, with equal probability, have live both in lacustrine and riverine habitats. In several cases (MOTUs A, F and G), the extant riverine populations derived most probably directly from lacustrine ancestors. In case of MOTUs A and G the

lacustrine ancestors of riverine populations were of Pleistocene age, while in the case of MOTU G, the presumed ancestor lived in the late Miocene.

Projection of the reconstructed longitudes and latitudes on paleomaps (Figs. 4B–4D) suggests that the early MRCA of extant *G. roeselii* MOTUs occurred in the area covered by the Neogene lake system occupying large parts of the western Balkan Peninsula. The reconstructed positions of the Miocene/Pliocene regional diversifications coincide with the major orogenic uplifts in the area, causing, among others, a decline of the lake systems and complete rearrangement of the local hydrological area.

### 28S: diversity, phylogeny and evidence for introgression?

For each of the 13 MOTUs identified by the COI data, one to 18 individuals (50 in total) were sequenced for the nuclear 28S marker, producing 23 haplotypes (N1–N23) (Table 1). The topology of the Neighbor-Joining tree (Fig. 5) showing relationships between the 28S haplotypes was partially concordant with the results derived from the mtDNA COI marker. For example, MOTUs with high COI divergence (ca > 15% K2p distance) were also divergent in the case of 28S. Only MOTU E showed a peculiar pattern as its individuals, associated with three 28S haplotypes (N2, N4 and N5), and belonged to two highly divergent clades of the 28S N–J tree (Fig. 5). Interestingly, MOTU E was always sympatric (sites 11, 13, 20) with another distantly related MOTU G, J and A, respectively. In two cases the individuals with COI haplotypes of MOTU E had a nuclear haplotype either close (site 11, N2 vs N3 and N9) or even identical (site 13, N4 and N5) to the nuclear haplotypes associated with the other MOTU present at the respective site.

### COI: diversity and demography within MOTUs

Analysis of genetic diversity within all 13 defined MOTUs (Fig. 6), revealed a generally rather high level of haplotypic and nucleotide diversity as well as K2p distance from 0.002 to 0.029 (Table 2). This was also reflected by the fact that in numerous cases various species delimitation methods have subdivided the MOTUs we defined into smaller units. In all cases this intra-MOTU divergence is younger than 2.5 Ma. Such sub-MOTUs have very localized distributions, e.g., they are restricted to one lake (e.g., Prespa, Vegoritis/Petron) or one lake system (Kastoria, Ohrid). This suggests that the Pleistocene glaciations might have promoted diversification at the local scale. Only MOTU M showed a clear sign of demographic expansion (Table 2, Table S6). On the other side, results of AMOVA (Table 2), pairwise $F_{ST}$ and Exact Test of Sample Differentiation (ETSD) revealed significant differentiation between most populations of the same MOTUs inhabiting various locations (Table S7).

## DISCUSSION

### High level and pre-Pleistocene onset of cryptic diversity

Our findings point out the presence of 12–29 different MOTUs, depending on the delimitation methods, that may represent plausible species. The higher estimates by GMYC and bPTP. which are methods based on phylogeny and coalescence, may be associated with their general tendency to overestimate the number of MOTUs in the presence of

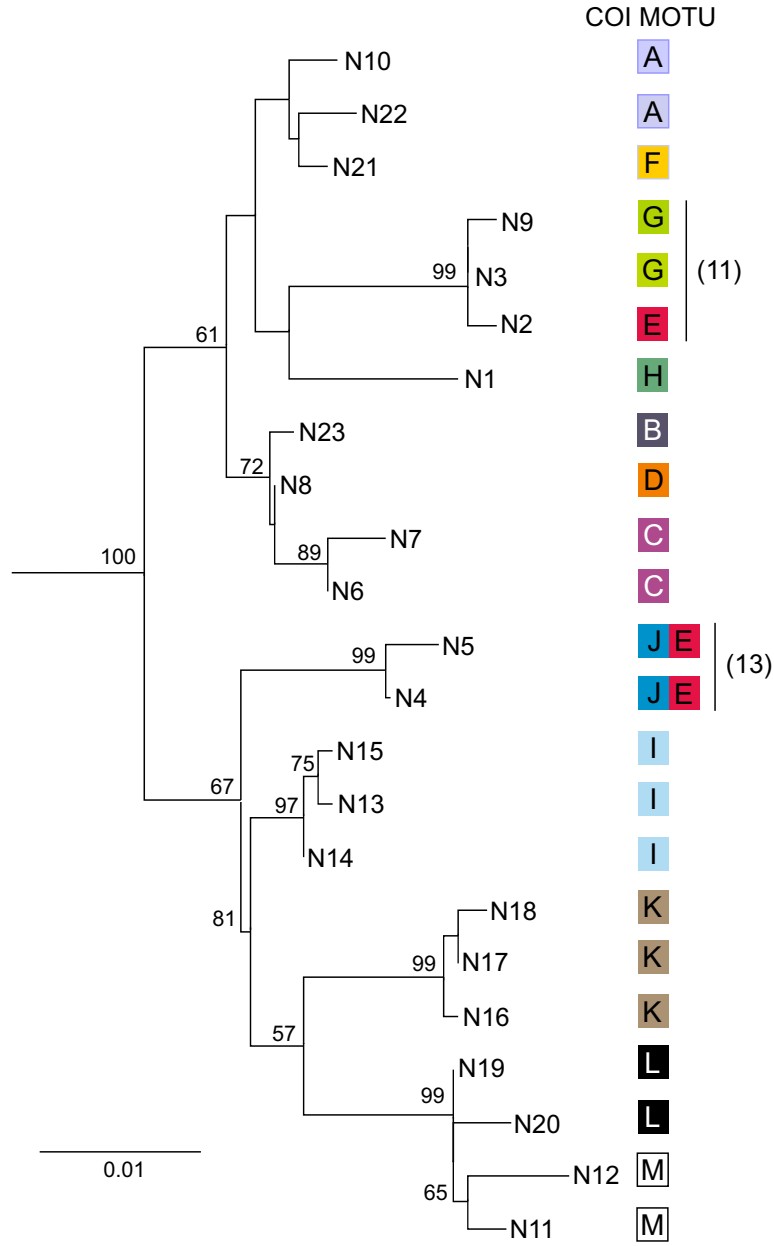

**Figure 5  Phylogenetic Maximum Likelihood tree based on uncorrected p-distance of nuclear 28S rDNA haplotypes (N1–N23), ascribed to COI MOTUs (A–M).** Color code for COI MOTUs is as in Fig. 2. Sampling sites are presented on Fig. 1 and in Table 1. Numbers in parentheses indicate sites where introgression between MOTUs was detected.

numerous highly divergent haplotypes, or when MOTUs appear to be unevenly sampled; with some containing low, and others high, within-MOTU genetic variability (*Talavera, Dincă & Vila, 2013*; *Zhang et al., 2013*). On the other hand, aggregating the haplotypes into BINs is based on a genetic distance threshold of 2.2% (*Ratnasingham & Hebert, 2013*) that may be too low for *Gammarus* (for details see *Lagrue et al., 2014*). Thus, it seems that the most conservative results of the ABGD analysis provide more realistic estimates. A detailed

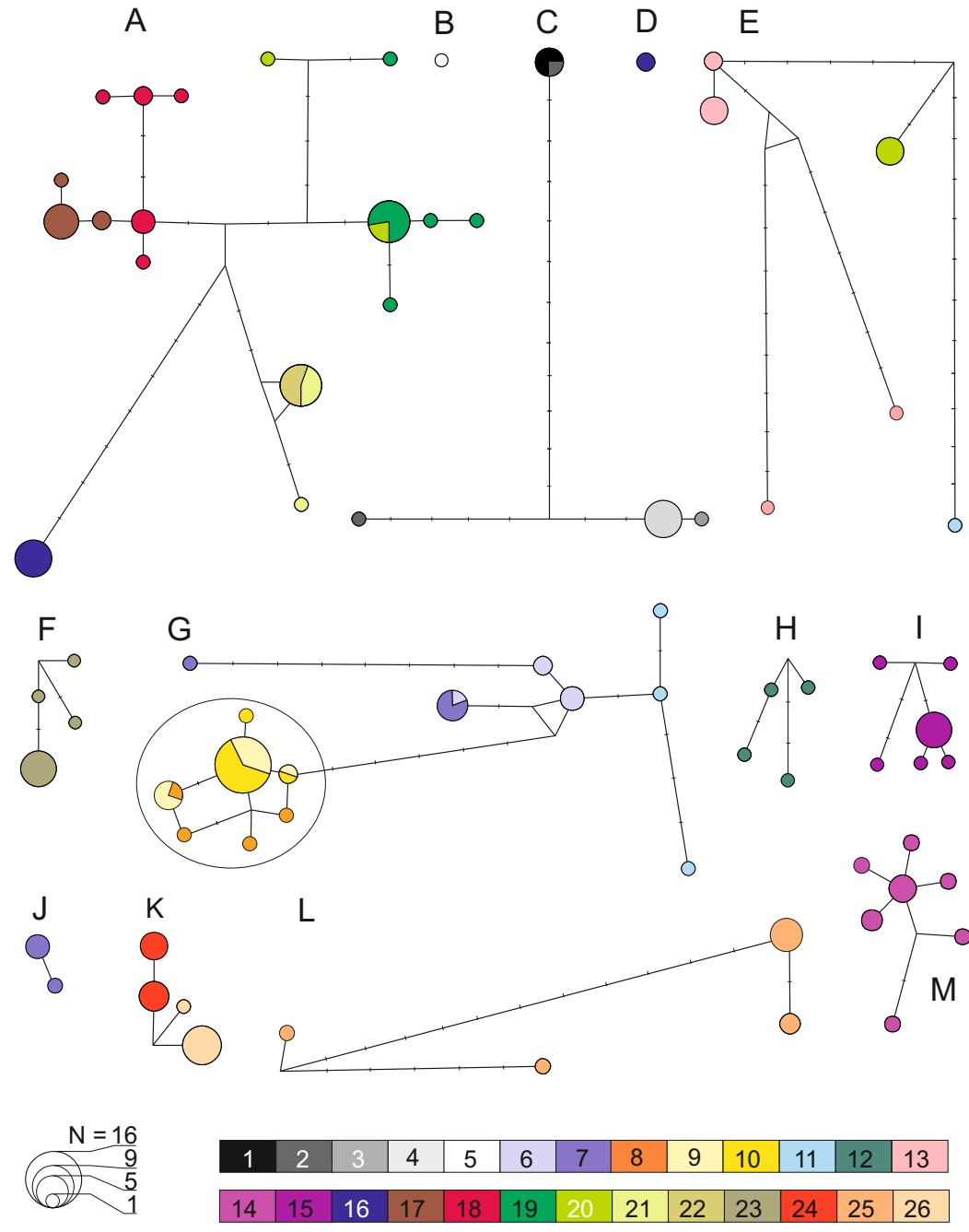

**Figure 6  Median joining network for COI within the 13 defined pre-Pleistocene MOTUs (A–M).** Each circle represent one haplotype with the surface being proportional to the number of individuals sharing that haplotype. Minimum distance between two haplotype represents a one nucleotide difference distance. A dash indicates a substitution. Node indicates potential ancestral haplotype. Colors are referring to the 26 sampled sites (details in Table 1).

**Table 2   Molecular genetic diversity, divergence, differentiation and historical demography based on mtDNA COI haplotypes in the MOTUs detected within *Gammarus roeselii* in the Balkans.** Locations, see Table 1 and Fig. 1B for details about locations. $N$, sample size. Diversity; k, number of haplotypes. $h$ et $\pi$, haplotype and nucleotide diversity, respectively. Divergence: K2p, mean Kimura 2 parameters phenetic distance between haplotypes with a given MOTU. Differentiation, overall $F_{ST}$ value. Demography: D, Tajima's $D$ and $Fs$ = Fu's $Fs$ tests.

| MOTUs | Locations | $N$ | Diversity | | | Divergence | Differentiation | Demography | |
|---|---|---|---|---|---|---|---|---|---|
| | | | k | $h$ | $\pi$ | K2p | $F_{ST}$ | D | Fs |
| A | 16[a] 17 18 19 20 21 22 | 49 | *18* | 0.90 | 0.0153 | 0.016 | 0.79** | $-0.08^{ns}$ | $-0.69^{ns}$ |
| B | 5 | 1 | 1 | 0 | 0 | n–c | – | 0 | n–c |
| C | 1 2 3 4 | 13 | 4 | 0.65 | 0.0151 | 0.021 | 0.45** | $0.78^{ns}$ | $6.09^{ns}$ |
| D | 16 | 2 | 1 | 0 | 0 | n–c | – | 0 | n–c |
| E | 11 **13 20** | 13 | 6 | 0.83 | 0.0189 | 0.029 | 0.68** | $-0.98^{ns}$ | $3.62^{ns}$ |
| F | 23 | 11 | 4 | 0.49 | 0.0033 | 0.006 | – | $-0.59^{ns}$ | $0.46^{ns}$ |
| G | 6 7 8 9 10 11 | 40 | 14 | 0.82 | 0.0112 | 0.016 | 0.76** | $-0.32^{ns}$ | $-0.56^{ns}$ |
| H | 12 | 4 | 4 | 1.00 | 0.0066 | 0.007 | – | $0.67^{ns}$ | $-1.01^{ns}$ |
| I | 15 | 12 | 6 | 0.68 | 0.0036 | 0.006 | – | $-1.05^{ns}$ | $-1.39^{ns}$ |
| J | 13 | 4 | 2 | 0.50 | 0.0009 | 0.002 | – | $-0.61^{ns}$ | $0.17^{ns}$ |
| K | **24 26** | 18 | 4 | 0.71 | 0.0028 | 0.004 | 0.79** | $0.94^{ns}$ | $0.89^{ns}$ |
| L | 25 | 9 | 4 | 0.69 | 0.0132 | 0.024 | – | $-0.46^{ns}$ | $3.65^{ns}$ |
| M | 14 | 10 | 7 | 0.91 | 0.0036 | 0.005 | – | $-1.40^{ns}$ | $-3.42^{*}$ |

**Notes.**

[a] sites in bold were part of the overall $F_{ST}$ estimate analysis.

Some sites were discarded as their sampling size were too small. Significance values ($p$) of the parameters were evaluated with 1,000 simulations: ns, not significant; $^{*}P < 0.05$; $^{**}P < 0.001$.

study upon the taxonomy of *Gammarus roeselii* is beyond the scope of our paper. In the only taxonomic review of European *Gammarus* made so far, and based exclusively on morphological characters, *Karaman & Pinkster (1977a)*, *Karaman & Pinkster (1977b)* and *Karaman & Pinkster (1987)* synonymised nine species and subspecies and recognised them as populations of *G. roeselii*. Our results are in clear opposition to their conclusion and suggest that, in reality, *G. roeselii* is a conglomerate of several locally endemic species. For example, our study revealed that MOTUs A and D were found in the Treska River of the Vardar system, while MOTUs A and E were found in the Struma River. Then, it is quite likely that at least some previously synonymised species, such as *Gammarus vardarensis* (Karaman, 1929), originally described from the Vardar River system, and *Gammarus strumicae* (Karaman, 1974), described from the Struma River system, correspond to some of these MOTUs. This points that another revision of *G. roeselii* should be made in the future, implementing the integrative taxonomy approach, which is the only reasonable option in the case of diverse animal groups with conservative morphology or poorly defined morphological variability (*Pante, Schoelinck & Puillandre, 2015*). High cryptic diversity with numerous divergent, locally endemic MOTUs representing possibly cryptic and/or yet undescribed species within other European freshwater amphipods was already evidenced for the morphospecies *G. balcanicus*, which is widely spread from the eastern Carpathians through the Balkan Peninsula and to the eastern Alps (*Mamos, 2015*; *Copilaș-Ciocianu & Petrusek, 2017*). Accordingly, a high level of cryptic diversity was found in case

of *G. fossarum* inhabiting Carpathians, Balkans, Alps and other upland areas in Europe (*Westram et al., 2011*; *Weiss et al., 2014*; *Copilaș-Ciocianu & Petrusek, 2017*).

Mountainous areas with steep elevational gradients, such as the Balkan Peninsula, have often been reported as local endemism and diversity hotspots of biota (*Hou, Li & Li, 2014*; *Luebert & Muller, 2015*; *Zhou et al., 2015*; *Katouzian et al., 2016*). In southern Europe it was often a result of speciation by isolation in consequence of severe alterations and fragmentations of habitats during Pleistocene climatic oscillations (*Hewitt, 2004*; *Salicini, Ibanez & Juste, 2013*; *Goncalves et al., 2015*). However, the presence of pre-Pleistocene lineages within conventionally recognised morphospecies was reported for several terrestrial and fewer freshwater taxa, including: reptiles; amphibians; and fish, but also insects and molluscs (e.g., *Albrecht et al., 2007*; *Marková et al., 2010*; *Falniowski et al., 2012*; *Garcia Munoz et al., 2014*; *Previšić et al., 2014*; *Pabijan et al., 2015*).

Our study revealed that the level of genetic divergence between most of the *Gammarus roeselii* lineages in the Balkan Peninsula is too high to be attributed to climatic fluctuations in the Pleistocene. Similarly, *Mamos et al. (2016)* as well as *Copilaș-Ciocianu & Petrusek (2017)* found that the divergence within *G. balcanicus* started some 15–20 Ma, in early/middle Miocene. *Copilaș-Ciocianu & Petrusek (2015)* found that *G. fossarum* diversified in the south-western Carpathians 10–17 Ma. A high cryptic diversity of pre-Pleistocene origin was also recently discovered within the freshwater isopod, *Asellus aquaticus* (Linnaeus, 1758), widely distributed in Europe including the Balkan Peninsula. The MOTUs of *A. aquaticus* diverged over Miocene and a number of them are localised in various parts of the Balkan Peninsula (*Verovnik, Sket & Trontelj, 2005*; *Sworobowicz et al., 2015*), which is similar to the spatial distribution pattern we have obtained for *G. roeselii*.

## The impact of Neogene and Quaternary paleogeography on the diversification of *Gammarus roeselii*

Historical framework of cryptic diversity within the Balkan freshwater taxa has been poorly studied, with a clear exception of taxa inhabiting ancient lakes, such as: the Ohrid; Prespa; or Trichonis (*Albrecht et al., 2008*; *Wysocka et al., 2013*; *Wysocka et al., 2014*). This is probably a result of the dynamic paleogeography and geology of the area, which may restrain a historical interpretation of phylogenetic relationships between MOTUs and their observed spatial distribution patterns, making it an uneasy endeavour.

Our BI chronograms based on two different calibration schemes provided estimations of mean COI substitution rates, in substitutions per site, per million years, which did not differ substantially (0.0113; SD 0.0016 vs 0.0129; SD 0.0030). They were also quite close to the COI mutation rate (0.0165; SD 0.0018) obtained for the cryptic lineages of *G. balcanicus* by *Copilaș-Ciocianu & Petrusek (2017)* and to the rate (0.0115) routinely used for arthropods (*Brower, 1994*). Thus, we can assume that our estimation of divergence times were reasonable and reliable in comparison to other studies. According to our results, the divergence of *G. roeselii* MOTUs took place within similar time frame as obtained in the other studies (reported above) upon *Gammarus*. The two main groups of MOTUs split at ca. 18 Ma (early middle Miocene) in the Neogene, and this continued their diversification to individual MOTUs until less than 1 Ma (Pleistocene) in the Quaternary. During this

time frame, the region was already a part of continental Europe; undergoing rapid changes of terrain due to the uplift of the Dinaric and Hellenic mountain chains. This included high tectonic activity of the area and recurrent reorganisations of local hydrological networks (*Popov et al., 2004*; *Dumurdzanov, Serafimovski & Burchfiel, 2005*). In the case of other gammarids (*Copilaş-Ciocianu & Petrusek, 2015*; *Copilaş-Ciocianu & Petrusek, 2017*; *Mamos et al., 2016*), we can assume that such landscape changes driven by Alpine orogenesis had a profound impact on the diversification of *G. roeselii* in the Balkans. A major feature of the Balkan landscape was large freshwater lacustrine systems, well supported by sediment and fossil data, existing throughout the Neogene and undergoing dynamic changes due to local orogenic movements (*Harzhauser & Mandic, 2008*; *Krstić, Savić & Gordana, 2012*). Unfortunately, we could not unambiguously define the ancestral habitats of Miocene common ancestors for the extant MOTUs; they could have lived in either rivers or lakes. However, reconstruction of the geographical origin of the Miocene and Pliocene ancestors suggests that they lived most probably in the areas occupied contemporarily by large freshwater lakes (*Popov et al., 2004*; *Harzhauser & Mandic, 2008*; *Krstić, Savić & Gordana, 2012*). This allows us to speculate that they could have been lacustrine organisms that diversified gradually along with the regression of the lakes and formation of new hydrological networks.

At least some of the extant ancient, or presumably ancient, Balkan lakes are probably relics of these paleolake systems (*Albrecht & Wilke, 2008*). Populations of *G. roeselii* were found in several of them, such as: Ohrid; Prespa; Trichonis etc. (see Table 1, Fig. 1). The lacustrine populations showed various levels of divergence from their sister riverine counterparts, dating from as early as ca. 10 Ma (Miocene) to as late as ca. 1 Ma (Pleistocene). In a couple of cases, populations of both types shared the same or closely related haplotypes. Our results for the ancestral habitat reconstruction suggest that for MOTUs A, F and G, the extant riverine populations derived from lacustrine ancestors. For example, the closest relatives of haplotypes found in rivers of Northern Greece (MOTU A) are those from the local lakes of Kastoria, Petron and Vegoritis, which are connected to the Aliakmonas River system. Similarly, in MOTU G, haplotypes from the Shkumbin and Drin rivers in Albania are most closely related to those from the ancient lakes Ohrid and Prespa. Some individuals from the Ohrid Lake share haplotypes with those from the River Drin. The lake drains to Drin and, possibly, was historically connected to the nearby Shkumbin River. Interestingly, for haplotypes from the currently isolated Lake Prespa, they form a sister clade to all the other haplotypes of MOTU G, from which they diverged 1.5 Ma. Similar sister relationships are also known e.g., within the gastropod genus *Radix* inhabiting both lakes (*Albrecht et al., 2008*). It is also known that Lake Ohrid and Prespa were at once part of the large Dessarete lake system (*Albrecht & Wilke, 2008*). The most recent studies of the lake sediment cores indicate that Lake Ohrid is at least 2 My old (*Jovanovska et al., 2015*). The age of Lake Prespa remains unknown but it may be at least as old (*Karaman, 1971*; *Radoman, 1985*). A split of that clade may reflect a separation of the Dessarete lakes and a relict character of the population inhabiting rivers derived from the Dessaretes. An interesting case is whether MOTU I is endemic to the ancient Lake Trichonis. This lake harbours several endemic species of molluscs and fish (*Doadrio & Carmona, 1998*;

*Albrecht et al., 2009*; *Almada et al., 2009*). Its exact age is not known, yet it is hypothesized to originate in the Pliocene, at ca. 3.4–1.8 Ma, or in the early Pleistocene (*Economidis & Miller, 1990*; *Khondkarian, Paramonova & Shcherba, 2004*). The Pleistocene origin of MOTU I diversity in the lake is supported by our results (Fig. S1, Table S3). Yet it is worth noting that it diverged from its sister, riverine MOTU J, at ca. 10 Ma. Given the presumably Pliocene/Pleistocene age of Lake Trichonis, and results of our ancestral state analysis, we may assume that MOTU I could derive either from a riverine or lacustrine ancestor. In fact, almost all the Miocene ancestors in our analysis had a rather equal chance to be either lacustrine or riverine. This may suggest that multiple habitat switches are evident in the evolutionary past of *G. roeselii*, and this points out the potential importance of lacustrine systems for their diversification history. Climate aridisation that had started in the Miocene and intensified greatly during the Pleistocene glaciations led to a periodical depletion of the riverine network in southern Europe, as seen in the present day genetic diversity of freshwater biota (*Tzedakis, 2007*; *Macklin, Lewin & Woodward, 2012*; *Gonzalez, Pedraza-Lara & Doadrio, 2014*). In such climatic conditions, the lakes could serve as local microrefugia, working as sources for the recolonisation of river systems, restored during more humid periods. That is particularly plausible for the Pleistocene, when recurrent glaciations resulting in droughts intermingled with humid interglacials. Indeed, our results indicate that the direct lacustrine ancestors of extant riverine populations for MOTUs A and G are of Pleistocene age. Similar conclusions may be derived from the results of the $F_{ST}$ and ETDS analyses showing significant differentiation between localities. Also, the aggregation of haplotypes into BINs may provide support for the Pleistocene speciation by an isolation process, as observed in other taxa (*Salicini, Ibanez & Juste, 2013*; *Goncalves et al., 2015*). It seems also that at least some MOTUs are versatile in changing habitats. They are present in both lakes and rivers, and we may also assume that changing between such habitats played an important role in the evolutionary history of *G. roeselii*.

## Evidence of potential introgression between currently divergent MOTUs

In several examples, we observed incongruence between phylogenies based on the nuclear (28S) and mitochondrial (COI) markers. Similar conflict between nuclear and mitochondrial genetic diversity patterns was evident for the Balkan cyprinid fish, suggesting an exchange of genes after initial isolation (*Marková et al., 2010*). Nuclear haplotypes shared randomly among numerous mitochondrial lineages were observed for *A. aquaticus* and interpreted as a result of incomplete lineage sorting (*Verovnik, Sket & Trontelj, 2005*; *Sworobowicz et al., 2015*). However, in the case of *G. roeselii* such interpretation is implausible. Two nuclear haplotypes were shared only between MOTUs E and J, which were not direct relatives and diverged from each other ca. 18 Ma. A third nuclear haplotype of MOTU E was a close relative of a haplotype from MOTU G; while, based on the COI sequence, the two MOTUs belonged to different clades that had separated ca. 8 Ma ago. Moreover such a situation was observed only in the localities where the MOTUs occurred in sympatry; suggesting that the female harboring nuclear haplotypes of MOTU E were fertilised by males from the other locally present MOTUs.

According to *Morando et al. (2004)* such phylogenetic patterns indicate secondary contact, resulting in hybridisation and/or introgression. It is hard to speculate what enabled such secondary contacts in the case of *G. roeselii*. MOTU E emerged at ca. 3.5 Ma, already in the Pliocene, and is the most diversified and internally divergent (five COI BINs) of all the defined MOTUs, and is also present in three distant localities. Thus, two main scenarios would be possible. One would assume a much wider presence of the MOTU before the Quaternary as well as an extinction event for much of the former distribution area due to Pleistocene climate aridisation resulting in a depletion of the river systems. According to this scenario, secondary contact with other MOTUs would be possible via recurrently changing watersheds during contemporary climate oscillations. A second scenario would assume a possible overland transport by birds. Such possibility was proved for freshwater amphipods living in lakes and shallow lowland rivers occupied by waterfowl (e.g., *Segerstråle, 1954*; *Rachalewski et al., 2013*). However it cannot be excluded, we believe it is less probable given that MOTU E was found exclusively in rivers in rather mountainous terrain.

## CONCLUSIONS

In conclusion, our study evidences that an old Neogene divergence of lineages resulting in substantial cryptic diversity may be a common phenomenon, if not a rule, in extant freshwater benthic crustaceans, such as gammarids, occupying areas that were not glaciated during the Pleistocene. Our study is among the first proposing a historical scenario behind the cryptic diversity observed in obligatory freshwater invertebrates. It highlights the role of Neogene lakes and development of associated hydrological networks in speciation mechanisms, and high level of local endemism, leading to intricate distribution patterns of particular lineages. Also, it points out that the ancient lakes of the Balkan Peninsula might have acted as local refugia during the drought periods, particularly in the Pleistocene. Finally, by evidencing the introgression events, we may suggest that *G. roeselii* is a complex of species at various levels of speciation.

## ACKNOWLEDGEMENTS

The authors wish to acknowledge several colleagues providing great help and company during the fieldwork: Janusz Hejduk, Radomir Jaskuła, Yannis Karaouzas, Piotr Spychalski, and Krešimir Žganec. We would also like to express our greatest thanks to all the local people in Albania, Bulgaria, FYR of Macedonia, Greece and Romania that have always treated us with enormous hospitality and were always willing to help. Without their selfless assistance we would never be able to succeed in gathering all the material or, in some cases, survive. A few additional samples from Slovenia, Croatia and Serbia were provided by courtesy of Prof. Dr. Boris Sket, University of Ljubljana, Slovenia. We are also very grateful to Dr. Kenneth De Baets, the Academic Editor of PeerJ, and to three anonymous reviewers, for their in-depth reviews and critical comments that allowed us to improve the quality of our manuscript. Mr Jamie Bojko from the University of Leeds, UK, kindly agreed to perform the language corrections. Finally, as music is an inevitable part of human life, we

would like to thank the young British music band London Grammar for their beautiful compositions that inspired us during data analysis and writing this paper.

### Funding
The sampling was performed during the Amphi-Balkan Expeditions II–V organised by the senior author and financed partially from the internal funds of the University of Lodz. This study was financially supported by the Polish Ministry of Science and Education, grant number N N303 579439. The sampling was performed during the Amphi-Balkan Expeditions II–V organised by the senior author and financed partially from the internal funds of the University of Lodz. The funders had no role in study design, data collection and analysis, decision to publish, or preparation of the manuscript.

### Grant Disclosures
The following grant information was disclosed by the authors:
University of Lodz.
Polish Ministry of Science and Education: N N303 579439.

### Competing Interests
The authors declare there are no competing interests.

### Author Contributions
- Michał Grabowski conceived and designed the experiments, analyzed the data, contributed reagents/materials/analysis tools, wrote the paper, prepared figures and/or tables, reviewed drafts of the paper.
- Tomasz Mamos conceived and designed the experiments, performed the experiments, analyzed the data, wrote the paper, prepared figures and/or tables, reviewed drafts of the paper.
- Karolina Bącela-Spychalska conceived and designed the experiments, performed the experiments, analyzed the data, contributed reagents/materials/analysis tools, prepared figures and/or tables, reviewed drafts of the paper.
- Tomasz Rewicz conceived and designed the experiments, performed the experiments, prepared figures and/or tables, reviewed drafts of the paper.
- Remi A. Wattier conceived and designed the experiments, analyzed the data, contributed reagents/materials/analysis tools, wrote the paper, reviewed drafts of the paper.

### DNA Deposition
The following information was supplied regarding the deposition of DNA sequences:
GenBank Accession numbers for mt COI: KP789673–KP789746.
Accession numbers for 28S rDNA: KP789747–KP789769.

### Data Availability
The raw data has been supplied as a Supplementary File.

## Supplemental Information

Supplemental information for this article can be found online at http://dx.doi.org/10.7717/peerj.3016#supplemental-information.

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

C-Q, Peng S-L. 2015.** Biodiversity of Jinggangshan Mountain: the importance of
topography and geographical location in supporting higher biodiversity. *PLOS ONE*
**10(3)**: Article e0120208.