# Peer review of "Neogene paleogeography provides context for understanding the origin and spatial distribution of cryptic diversity in a widespread Balkan freshwater amphipod"

_PeerJ, doi:10.7717/peerj.3016_

## Round 0.1 · original submission · Major Revisions

All 3 reviewers agree that your study is important and well worth publishing (extensive field work, good sampling and high quality data). However, the reviewers raised several issues that need to be addressed before acceptance. The main points are:

1) Use of COIs vs. 28S: Why did you not analyze 28S and COIs together (see comments by reviewer 2) ?

2) MOTUs: There seems to be some issues how you define/use MOTUs and how you interpret your results (see comments by reviewer 2 and 3)

The reviewers were quite thorough so i do not have much to add.
In addition to the suggestions of the reviewers, please also address the following:

• Secondary calibration: as far as I understand you used secondary calibration to constrain your study and compare it with other studies– there might be some issues with this approach (you might be propagating calibration issues); Did the other studies you are comparing with use the same calibration?

• Geological age: you discuss age reported in geological literature for change in paleogeography (paleolake configurations) and glaciations; it would be necessary to state on which kind of data (direct dating or rather stratigraphic dating). In the latter case, ages might shifts depending on correlations and recalibrations of the geological time-scale. Furthermore, different studies might have different interpretations. The potential influence of these differences should at least be discussed somewhere.

Reviewer 1 ·

Basic reporting

The basic reporting is excellent

Experimental design

reasonable

Validity of the findings

Yes

Additional comments

Comments to the Author
"Between lakes and rivers - Neogene paleogeography provides context for understanding origin and spatial pattern of cryptic diversity in a widespread Balkan freshwater amphipod"

The manuscript presents very interesting results of Gammarus diversification associated with gradual deterioration of palaeolakes and reveals 13 cryptic species of Miocene origin.

I recommend to publish this manuscript with corrections.
1) please note the use of n-dash and m-dash. "10-17 Ma" should be "10–17 Ma". Please look out it throughout the text.

2) In the section of Result, the substitution rate was calculated as 0.0113. I think in the section of Discussion, the authors should give some comparison for this rate to confirm that it is reasonable.

3) Please give topics for the section of Discussion.

Reviewer 2 ·

Basic reporting

No comments, all criteria requred are met.

Experimental design

I have few comments related to methodology, see "General Comments".

Validity of the findings

No Comments.

Additional comments

The manuscript describes extensive yet overlooked speciation within G. roesseli species complex, and attempts to link it with paleogeologcal events. Overall this is a sound study, and I recommend its acceptance after revision. The strength of the study is massive field work, a thorough sampling and high quality data. A possible weakness may be some analyses, I feel the MS would benefit if authors explained the logic of analytic procedure more in detail. More specifically:

Introduction
Introduction section is well written, clear and well presents the problem with all relevant literature. Hypotheses are clearly stated, however, I suggest a reformulation of the second hypothesis. Authors state:
Lns 147-148: Second, we hypothesise that there will be a deep-level divergence between the delimited MOTUs, reaching far beyond the Pleistocene Ice Ages.

One part of the problem is that this hypothesis is tautological, i.e. if the divergence was not deep enough, the MOTUs would not be detected with methods applied. The second part of the problem appears in results section, where authors define MOTUs which are proposed by ABGD and are old enough to be considered as MOTUs. I suggest simplification of the hypothesis eg that most of diversification and speciation events preceded Pleistocene. This is easily testable with LTT plot.

Materials and Methods:

Species delimitation methods, tough unilocus alone, were performed well and carefully. Yet, in this section I have few questions, some of them may be beyond the scope of the study, some of them may authors easily clarify with some additional explanation, or they may decide to invest more time into analyses.

The first question I asked myself is how authors know that this species complex is monophyletic. As they showed in a previous study, monophyly of cryptic species complex should not be assumed per se. Since they analyze range shifts and ecological diversification, non-monophyly may influence the final conclusions. I might have overlooked this, but as a reader, I would be grateful if I found a statement the complex is monophyletic, or a broader phylogenetic analysis in which this monophyly was shown.

My second question is why authors did not perform phylogenetic analysis using 28S fragment, or joint analysis of COI+28S fragments. It is a bit impractical to compare very different presentations of the data; moreover, hierarchic relationships among lineages might be different and consequently affect reconstruction analyses. I think that COI and 28 in this study are enough congruent to warrant concatenated analysis.

My third question relates to reconstructions of ecology. As I could see from Fig. 2, reconstruction of ancestral states (lake, riverine) is based on data for individuals but not MOTUS. The MOTUs, the units of interest, are hence not equally represented in the analysis; higher number of individuals may unintentionally increase the impact of a species. This is particularly evident in clade A-E. All species, except A seem to be riverine, except MOTU A which is polymorphic. With other words, this species does not care whether it lives in streams or lakes. In the analysis are included 9 individuals from lakes (all MOTU A), whereas the entire clade counts 29 individuals (MOTUs A-E).
All 9 individuals belong to polymorphic MOTU A and apparently strongly influence the reconstruction. If authors repeated the analysis using 5 tips only (MOTUs A-E), of which 4 tips were lake species and one was polymorphic, I am almost sure they may get different results.

My fourth question goes to validity of reconstruction of ancestral range. The authors performed similar analysis as Mcinnerney et al. (2014), yet I am afraid that the method they used is not appropriate (I may be wrong). The problem of reconstruction of ancestral ranges is that reconstruction needs to incorporate two processes, namely vicariance and dispersal. If vicariance plays a major role (and authors suggest it is crucial for understanding amphipod history), ‘evolutionary’ changes happen almost instantaneously, when ancestral population is split into two or more daughter populations. By contrast, reconstruction models of species traits consider evolutionary changes more or less gradual, which take place along the branch according to selected evolutionary model (Brownian motion, Orstein-Uehlenback). In present study, authors treated geographic coordinates as quantitative traits and reconstructed them as heritable traits. Although I think the result will not change dramatically, I would be more comfortable if authors repeated analyses using some of more appropriate software (many programs are implemented into RASP http://mnh.scu.edu.cn/soft/blog/RASP/ ).
Moreover, I wonder whether or not the results were more informative, if they were performed on a coarser grain, e.g at the level of drainage system. I leave decision to authors.

Results

Results are well represented; I only have some minor comments:
Lns 313-314 – merely repeats the sentence from MM section
Ln 330: Authors report SE in brackets, but they do it explicitly only on the first occasion. I would add ‘SE = ‘ in all cases to make it clear.
Ln 382: ‘Diverge’ should be replaced with ‘Divergence’, right?
I miss some labels at figures. In Fig. 2 the nodes are labelled using two numbers – what they indicate?

Discussion:

In my opinion, the MS would benefit if the Discussion was shortened and more compact. There are three major conclusions, namely that speciation took place before plesistocene, that species structure needs to be revised and that the evolution of the complex somewhat corresponds to montane uplift and where the ancient lakes probably played an important role.

Minor comments:
Ln 517 : ABGD is not the most parsiomonious result (technically probably yes, but I doubt the authors refer to assumptions underlying the method). I would rather state ‘conservative’ or something similar.
Ln 566: Lake Ohrid and River Drin do not share haplotypes, but thier inhabitants may share haplotypes.

Reviewer 3 ·

Basic reporting

The authors combine molecular derived data with current habitat information to reconstruct the paleogeographical history of a widely distributed amphipod, Gammarus roeselii, in the Balkan peninsula. Given the observed degree of previously unknown cryptic diversity within this morphospecies, several MOTUs are reported and putative new species hypothesised. The manuscript adds substantial knowledge to the field of freshwater amphipod research in Europe and sheds light on possible processes involved in divergence and speciation within this ecologically important group.
The manuscript fulfils the PeerJ submission and publication requirements and is, from the reviewers side of view, suited to be published in this journal. Nevertheless, the manuscript needs substantial revision, particularly in language style and some analytical parts as well as in contingency of presenting.
The introduction provides a good overview about the Balkans as a biodiversity hotspot, interesting due to its geological past and suited to study processes of divergence and speciation driven by paleogeography. Nevertheless, the introduction should probably be restructured, starting with the phenomenon of cryptic diversity, especially in freshwater amphipods in Europe, followed by the possible explanation by paleogeography. By this, the ordering of hypotheses and results is more straightforward and meaningful as well. The clarity would profit. The background on cryptic diversity in freshwater amphipods could be extended to e.g. Gammarus fossarum as well as to niphargids, showing many of the reported phenomena. Overall, the structure is conform to the discipline norm. Some figure captions need improvement (see detailed comments). Raw data will be made available via GenBank, but a supplemental table with all sampled sites (coordinates, names etc.) would improve the provided data.

Experimental design

The study fits the scope of the journal. The scientific methods are sound and the analysis is appropriate to answer the posed questions. The presented work profits from the nice combination of phylogenetic reconstruction with a spatial interpretation. The combination of mitochondrial and nuclear DNA is corroborating the results. The authors state four hypotheses that they want to test in their study.
Unfortunately, they don't provide the reasoning why they expect their hypotheses to hold. Hence improvement of reasoning or citing literature is needed to support the expectations of high cryptic diversity within G. roeselii and the expected deep-level divergence. Otherwise the authors should state, that they were interested for these data, without antedecent expectations. For the fourth hypothesis of relating geological events to the diversification, the term "test" should be avoided since there is no formal statistical test conducted later in the manuscript.
Some methodological parts need improvement. The 2% pairwise distances for sequences within mtCOI as approximation for species delimitation (used in ABGD, derived from Costa et al., 2007; mostly focussing on Decapoda) is really low and was higher in other published research. The reviewer would be interested in the outcome of analysis given a higher value (e.g. Weiss et al., 2014, or Katouzian et al., 2016, found higher values between putative species). The approach to infer ancestral coordinates is poorly described and could not be checked for suitability by the reviewer due to lack of information (BayesTraits methods description needs improvement). The lineages through time plot is not meaningful when conducted on all available individuals, since the last lineage split happens about 3.3 Ma. The increased diversification rate is an artefact from plotting all haplotypes (not identical to lineages). In the Material and Methods part, three delimitation methods are described (ABGD, bPTP and GMYC), whereas in the Results section and in Figure 2, four delimitation methods are mentioned (GMYC is further split into a single and a mixed approach). This should be made consistent. Furthermore, the raw data could not be checked on GenBank since they are still embargoed (this point is not meant as improvement suggestion). The reviewer would suggest that the authors additionally make the data available via BOLD. Like this, the quality of sequences gets checked automatically and assignment to BINs happens automatically. This would provide an additional way of delimiting putative species.

Validity of the findings

The reported findings are conclusive and the solid data corroborate the high degree of cryptic diversity present in freshwater amphipods, particularly in Europe. As stated above, some methodological parts need improvement to support the findings in all details. The paleogeographical explanation is reasonable, but remains a bit speculative. This is not a negative point and the authors do not state otherwise. Unfortunately, the paper misses the chance to resolve the cryptic status of the reported MOTUs. The work is based solely on MOTU delimitation, not including morphological or deeper ecological analysis. Hence it is another freshwater amphipod paper stating high cryptic diversity without resolving it taxonomically (e.g. Müller et al., 2000, Westram et al., 2013, Weiss et al., 2014). For applied fields and conservation approaches this situation is rather unsatisfying. The cryptic status is not proven with morphometric data, as for example in a recent paper by Jaskuła et al., 2016, published in PeerJ as well (and with the first author of the manuscript as co-author). The main finding of a high degree of molecular diversity with several MOTUs is solid, and the proposed paleogeographical explanation is reasonable.

Additional comments

General comments:
- Language and style needs substantial improvement. For example there are many articles (either "the" or "a") missing.
- There seems no common agreement on the naming of the species. Many authors use G. roeseli, even though the World Amphipoda Database uses G. roeselii (as first proposed by Gervais in 1835). The redescription with a neotype by Karaman & Pinkster in 1977 uses G. roeseli. A Web of Science search reveals that a majority of published papers uses the epithet roeseli (112 vs. 40) when searching for "Gammarus roeseli" vs. "Gammarus roeselii". The reviewer is fine with both names, even though more used to G. roeseli.

Detailed comments:
- Title: "Between lakes and rivers" seems a bit forced and only becomes clear throughout the manuscript. If something catchy is wanted, I would rather go with a beginning as in one of the subheadings "A neogene tale - Paleogeography provides context ...". Further I suggest changing "pattern" to "distribution".
- Line 32: It was not really "tested" but rather hypothesised and looked at since there is no statistical test (with a null hypothesis).
Line 42: Move the part about 28S to the end of section (first all COI, then 28S).
Line 56: Use "relict" instead of "relic" throughout the manuscript.
Line 104: remove "just", as it is diminishing work done by other scientists.
Line 108: State Ma for the geological periods, so the reader knows what you mean with Neogene or Pleistocene. For each period, when mentioning it the first time.
Line 123: There are studies showing that amphipods may survive a prolonged period of drought and are transported by birds or humans (e.g. Segerstråle, 1954, or work by the authors themselves, 2013). Otherwise state the meaning of prolonged.
Line 136: Mention GERVAIS 1835 as author of G. roeselii.
Line 144: Start the goal of the study with "Aim is to study the paleogeographical influence on the diversity of a freshwater amphipod..." or restructure the introduction, starting with G. roeselii. Is the aim of the study on G. roeseli or on the paleogeographical influence on the diversity of a species?
Line 157: A list of all sampled sites as supporting information would be very nice and helping in understanding the distribution of G. roeselii.
Line 159: Correct citation to "Jażdżewski ..."
Line 173: State which primer was used for which individuals (or was it a mixture?).
Line 205: The models compared to should be stated. The MCMC method of moments estimator is suited to compare the models, but the reader misses the information about alternative models of sequence evolution.
Line 216: LTT (not LLT)
Line 222: State that the three methods were used in COI. The sentence is not structured well: There should be a full stop after "used", or it should be restructured to "..., the three methods used were i) a distance-based ...".
Line 223: Three methods are introduced, whereas later in the manuscript (line 353 and Fig. 2) the authors differentiate four different mehtods (single and multiple GMYC): adjust.
Line 234: The value of 2% intraspecific distance is not as conservative as stated in the manuscript. More recent amphipod literature reports higher values.
Line 274: To estimate ancestral coordinates (continuous data) in BayesTraits, a two-step process is needed (Manual: first a distribution of models is estimated from available data, secondly the models are used to estimate unknown values). Please explain this method in more detail, since reconstructing these coordinates is crucial but not trivial.
Line 287: Remove "A Pleistocene tale", since it is only methods section.
Line 290: Don't mention the number of MOTUs before results section.
Line 294: Given the substitution rates resulting from the BI analysis? Or are the weight applied ambiguously?
Line 297-309: Unclear paragraph. Please reformulate. Also, sample size might be to small to have meaningful results, see Fitzpatrick, 2009. “Power and Sample Size for Nested Analysis of Molecular Variance.” Molecular Ecology 18 (19): 3961–66. doi:10.1111/j.1365-294X.2009.04314.x.
Line 301: correct to "MOTUs A, C, E, G and K", "assuming" to "given" (or is it really just assuming?)
Line 302: Fst is written as F with subscript ST.
Line 317: correct m asl. to MASL or m.a.s.l.
Line 318: What are the coordinates (in °E) of "the Adriatic/Ionian coast"?
Line 319: Drava River is 46°N, right? Not E.
Line 320: "thorough" or "extensive", not "throughout"
Line 322: Bosnia & Herzegovina (not Hercegovina)
Line 348: The accumulation of lineages is an artifact of not comparing lineages but all haplotypes! Correct. Drawing only the putative 13 lineages would presumably not show an increase over time. Further the axis is not log-transformed, so it's not a log-lineages through time plot.
Line 382: Divergence, not Diverge.
Line 389: "diverged from M...", not "diverged from Ma...".
Line 401 & 410: MOTU J mentioned, but site description is different (tributary of Seman River vs. remote locality on the upper Devoll River). Adjust.
Line 411: state that MOTU K was found together with MOTU E. This is a mix of a northern and southern MOTU.
Line 431 & 466: These sentences should be part of the discussion.
Line 497: The reference is "Copilaş-Ciocianu & Petrusek (2015)"
Line 561: remove "in MOTU A", and add "(MOTU A)" after "Northern Greece".
Line 620ff: Add subtitle "Conclusions".
Line 628ff: Funding is not part of Acknowledgments in PeerJ.
Line 633: Capitalize Prof. and Dr.

Table 2: h diversity should be denoted with an italic h. k2p should be K2p. Again Fst should be F subscript ST.
Fig. 1 & 2: Depending on the print size, the font might be too small.
Fig. 2: Please state the meaning of the numbers at nodes. Adding the split age at nodes might help.
Fig. 3: This plot should be generated from MOTUs, not haplotypes (lineages, not individuals).
Fig. 4: remove "-" in front of "16-15 Ma" in b) plot description. Also, "All" in b) plot looks like AII and should be changed to "A-M" (more consistent).
Fig. 5: Description is insufficient. What are the distances? What do the lines and nodes mean? For better readability, remove the part after the "N" in the names. Just state the haplotype names (N1-N23).
Fig. 6: Again, what are the distances, lines, nodes?

---

## Round 0.2 · Minor Revisions

Thank you for integrating and addressing all the points raised by the reviewers and myself. There are still some minor points which need to be addressed:

English Language: The English language could be improved to ensure that your international audience can clearly understand the text. Some example where the language can be improved include lines 32, 45, 57, 87, 133, line 179-180, line 629 – current phrasing makes it difficult to understand what you mean in some cases. I therefore suggest to have a native English speaking colleague proofread your manuscript as I am not a native English speaker myself.

Definition of biological species: I agree with reviewer 2 that might be good to drop “biological” before species as you can only make inferences

Link between MOTUs and ancestral habitat: I also agree with reviewer 2 that it might be appropriate to highlight an alternative possibility that the link between environment and MOTU is less mandatory than discussed in present version. A single sentence stating this and potentially the reason why this might be less likely than alternative you follow in your study would suffice.

Figures: I agree with the reviewers that the manuscript and particularly the updated figures have considerably improved the flow and understanding of your results and interpretations. Figure 2 might still benefit further from adding the ranges of the geological epochs as well as the main geological events discussed in the text. Figure 5 might benefit from adding the different sites where these haplotypes were sampled to illustrate sympatry (as you discuss in the text)

LTT Plots: Maybe I have missed something, but aren´t you supposed to compare LTT plots with appropriate null models to make certain inferences/interpretations about rates?

In addition to these points and the comments by the reviewers, please also address the following additional points:

Line 26: I guess it should be “The Balkans” rather than Balkans (or alternatively the Balkan Peninsula)
Line 32: “preceded Pleistocene”; Do you mean preceding the Pleistocene ?
Line 45: “Neighbor Joining Tree was”; “The Neighbor Joining Tree approach” or “The Neighbor Joining Tree method” might be more correct.
Line 57: “Extant relict lakes” sounds odd; Do you mean recent / still present relict lakes / currently extant relict lakes
Line 84: “fortified”; “potentially intensified” might be more appropriate
Line 87: “recurrent river captures”; unclear; please rephrase
Line 99-101: Although this might be true, this is still an inference in each study; it would write something like “highlighted” or “revealed that it was probably a key glacial refugium”
Line 133: “exceptional” might be more appropriate than “peculiar”
Line 145: “20 – 15 Ma”; you fluently change between million years and geological epochs, but the reader might not be so appropriate with these – certainly not in the beginning of the manuscript so I suggest to add to which epoch (early to middle Miocene) this belongs for completeness sake
Line 160-161: Not really, you use the predicted or inferred paleogeographic distribution
Line 179-180: I think this should be “found at 26 sites”
Line 215: I think geological calibration points or geological constraints sounds more appropriate
Line 259: Are you supposed to compare the lineage through time (LTT) plots with null model to see if they deviate from your expectations?
Line 395: It might be interested to point out that the one with geological constraints is the most abberant and also has the highest standard error
Line 404: It might help to add the age in million years after Miocene and Pliocene
Line 506-508: It might be worthwhile to discuss the possible reason for the close or identical haplotypes
Line 517: As this is inferred, I suggest to write “might have promoted” instead of promoted
Line 539-541: It might be worthwhile to visualize these potential taxa on the graphs for completeness sake (if possible)
Line 564-565: Could the difference be related with using a different calibration ?
Line 598-591: It think it might help to add the ranges of these events to the chronogram for comparative purposes
Line 629: I think it should be “hypothesized”
Line 688-697: It would be appropriate to thank the reviewers in the Acknowledgements
Figure 2: the graph would benefit from adding the used geological time-scale to the chronogram as well as major geological events discussed in the text for comparative purposes
Figure 3: I think it would be appropriate to compare you LTT with null models
Figure 5: This figure might benefit from adding the different sites where these haplotypes were sampled to illustrate sympatry (as you discuss in the text)

I am looking forward to the revised manuscript - thanks again for sending this interesting manuscript and taking our suggestions at heart.

Reviewer 2 ·

Basic reporting

I have no comments.

Experimental design

I have no comments.

Validity of the findings

In general, I agree with authors. Yet, one part of the discussion (lns 572 onward) may be complemented with one statement. This text is very detailed discussion on the ecological origin of the MOTUs. I overlooked in a previous round of revision / I expected different results from revised version, but maybe authors should mention also an additional scenario, i.e. that all MOTUs simply do not care if they are lacustrine or riverine. In fact, the link between environment and MOTU is inferred from filed observation and may be less mandatory as discussed in present version. The authors hint into this direction when they consider lakes as microrefugia; this suggests that the barrier between flowing and stagnant water is not perfect.

Additional comments

Subheadings in line 208 and 258 both announce an analysis of diversification rates.
Ln 259: no need for indent paragraph.
ln 357: a letter ‘c’ after citation seems as a type-setting mistake.
Lns426-429: This sentence is rather long and difficult to read. Perhaps split it into two.
Ln 526: The authors state that the delimitation methods may unveil biological species. Biological species is based on reproductive barrier that can be in this study inferred only indirectly and not for all species pairs. I do not doubt into author’s conclusions, but I am well aware of some readers who may criticize such statement as unsupported. The critics may gain some support in putative introgression hypothesis. I would suggest that authors remove the term ’biological’; the message shall remain intact but less debatable.

Reviewer 3 ·

Basic reporting

The reviewer appreciates the efforts made by the authors and acknowledges the extensive response letter. All the points that were brought up were resolved and covered satisfactory. Apart from remaining minor issues with English language the reviewer recommends the manuscript for publication in PeerJ. The data present an interesting case study and will help to understand diversification processes, particularly in an ecologically relevant order of freshwater macroinvertebrates. The manuscript gives a good background and is well referenced. It is well structured and consistent. A point brought up by the reviewer was not covered (sampling sites coordinates as supplement are still not available), but the reviewer is fine with this and agrees on the answer. It is not important for the aim of the manuscript but would be relevant for methodological reasons. Nevertheless just site names and coordinates would be appreciated, without additional data. The restructuring of hypotheses improved the manuscript substantially and is now clear. The organization of the discussion greatly profited from the extensive changes and additions made by the authors. Including taxonomic references and suggesting resurrecting some synonymised taxa are valuable additions to the discussion and cover the remark on cryptic taxonomy made by the reviewer.

Experimental design

As with the previous version, all mandatory points of the experimental design are met and sufficiently documented. Some initial analytical issues were resolved.

Validity of the findings

Changing figures and including many small clarifications make the findings compelling and conclusive. The data base is solid, sampling effort high and extensive. The conclusions are clear and answer/confirm the initially stated hypotheses. Suggestions regarding the taxonomy are a welcomed addition.

Additional comments

Minor issues when reading through:
Line 151: Change "studied" to "studies"
Line 164: Make Gammarus in italics
Line 185: "sorted at site" (since not all at the same site)
Line 223: According to World Amphipoda Database, Dikerogammarus belongs to the Gammaridae family
Line 260: "a" or "the" before "tree"

---

## Round 0.3 · accepted · Accept

Thank you for integrating our final suggestions.